# Secondary Organic Aerosol Formation via Multiphase Reaction of Hydrocarbons in Urban Atmospheres Using the CAMx Model Integrated with the UNIPAR model

Zechen Yu[1,], Myoseon Jang[1], Soontae Kim[2], Kyuwon Son[2], Sanghee Han[1], Azad Madhu[1], and Jinsoo Park[3]

[1]Department of Environmental Engineering Sciences, Engineering School of Sustainable Infrastructure and Environment, University of Florida, Gainesville, FL, USA.
[2] Department of Environmental and Safety Engineering, Ajou University, Suwon, South Korea.
[3]Air Quality Research Division, National Institute of Environmental Research, Environmental Research Complex, Incheon, South Korea.

*Correspondence to*: Myoseon Jang (mjang@ufl.edu)

**Abstract.** The prediction of Secondary Organic Aerosol (SOA) in regional scales is traditionally performed by using gas-particle partitioning models. In the presence of inorganic salted wet aerosols, aqueous reactions of semivolatile organic compounds can also significantly contribute to SOA formation. The UNIfied Partitioning-Aerosol phase Reaction (UNIPAR) model utilizes the explicit gas mechanism to better predict SOA formation from multiphase reactions of hydrocarbons. In this work, the UNIPAR model was incorporated with the Comprehensive Air Quality Model with Extensions (CAMx) to predict the ambient concentration of organic matter (OM) in urban atmospheres during the Korean-United States Air Quality (2016 KORUS-AQ) campaign. The SOA mass predicted with the CAMx-UNIPAR model changed with varying levels of humidity and emissions and in turn, has the potential to improve the accuracy of OM simulations. The CAMx-UNIPAR model significantly improved the simulation of SOA formation under the wet condition, which often occurred during the KORUS-AQ campaign, through the consideration of aqueous reactions of reactive organic species and gas-aqueous partitioning. The contribution of aromatic SOA to total OM was significant during the low-level transport/haze period (24-31 May 2016) because aromatic oxygenated products are hydrophilic and reactive in aqueous aerosols. The OM mass predicted with the CAMx-UNIPAR model was compared with that predicted with the CAMx model integrated with the conventional two product model (SOAP). Based on estimated statistical parameters to predict OM mass, the performance of CAMx-UNIPAR was noticeably better than the conventional CAMx model although both SOA models underestimated OM compared to observed values, possibly due to missing precursor hydrocarbons such as sesquiterpenes, alkanes, and intermediate VOCs. The CAMx-UNIPAR model simulation suggested that in the urban areas of South Korea, terpene and anthropogenic emissions significantly contribute to SOA formation while isoprene SOA minimally impacts SOA formation.

## 1 Introduction

The formation of secondary organic aerosol (SOA) has gained substantial interest from researchers because of its important impact on climate change (Ipcc, 2015; Seinfeld and Pandis, 2016), urban visibility (Chen et al., 2012; Ren et al., 2018), and human health (Requia et al., 2018). In urban atmospheres, emissions from industries, fuel combustion, and motor vehicles are major contributors to the observed concentrations of organic aerosol via both primary emissions and SOA formation (Gentner et al., 2017).

SOA forms via traditional gas-particle partitioning (Odum et al., 1996) of oxygenated products formed from hydrocarbon (HC) photooxidation. Additionally, the heterogeneous chemistry of these oxidized carbons in the aerosol phase significantly contributes to SOA burdens (Hallquist et al., 2009; Kalberer et al., 2004; Tolocka et al., 2004). These reactions include condensation reactions between organic species (i.e., hemiacetal/acetal formation and aldol condensation) (Jang et al., 2002; Jang and Kamens, 2001; Tobias and Ziemann, 2000), reactions of organics with water (i.e., hydration of aldehydes and hydrolysis of epoxy products) (Jang et al., 2002; Hallquist et al., 2009; De Haan et al., 2009), and the formation of organonitrates (Farmer et al., 2010) and organosulfates (OS) (Surratt et al., 2007). The contribution of heterogeneous chemistry on SOA growth has a larger role in the presence of electrolytic wet aerosol (Volkamer et al., 2007; Volkamer et al., 2009). Polar organic species can dissolve in the aerosol aqueous phase induced by hygroscopic sulfate and increase SOA mass by forming nonvolatile oligomeric matter. Additionally, the aerosol acidity associated with wet inorganic sulfate can catalyze aqueous reactions of organic species (Jang and Kamens, 2001; Jang et al., 2002; Kuwata et al., 2013; Limbeck et al., 2003; Kleindienst et al., 2006; Lewandowski et al., 2015; Hallquist et al., 2009). Few models account for aqueous reactions of several products (i.e., glyoxal and IEPOX (epoxy diols form isoprene products)) that potentially may significantly impact SOA formation (Ervens et al., 2011; Sumner et al., 2014; Budisulistiorini et al., 2017; Knote et al., 2014). However, current models poorly integrate the multiphase chemistry of many other organics into SOA mass predictions. In particular, the model applied to regional scales suffers from a substantial negative bias under high humidity conditions (Heald et al., 2011; Pye et al., 2017; Li et al., 2020). Park et al. (2021) extensively evaluated the prediction of the organic aerosol produced during the KORUS-AQ campaign by using different air quality models, which were varying in chemistry mechanisms, aerosol thermodynamics, the types of SOA precursors, and the SOA schemes. In their study, the SOA formation was simulated with the SOAP, the 4 bin-base VBS or the 5-bin-base VBS modules. The predicted organic aerosol masses were, however, underestimated compared to observation data (HR-ToF-AMS) suggesting the limitation of the current SOA modules. The SOA model, such as the partitioning-based two product model, has no feature for SOA formation via aqueous phase reactions of different oxygenated products formed from various HCs.

The UNIfied Partitioning-Aerosol phase Reaction (UNIPAR) model was developed by Im et al. (2014) to predict SOA mass based on multiphase reactions of toluene and 1,3,5-trimethylbenzene, was developed. In the UNIPAR model, the products predicted using explicit gas mechanisms are lumped based on volatility and emerging chemistry in the aerosol phase. This UNIPAR model has been extended to various SOA originating from isoprene, terpenes, aromatics, and gasoline and

demonstrated through the extensive photochemical outdoor smog chamber data (Beardsley and Jang, 2016; Cao and Jang, 2010; Zhou et al., 2019; Yu et al., 2021b; Han and Jang, 2022).The model parameters linked to thermodynamic properties and aerosol chemistry are also estimated according to lumped species characteristics. By exploiting the explicit structures of oxygenated products, the UNIPAR model is capable of processing aerosol chemistry to simulate SOA formation at different phase states (i.e., dry and wet) of salted aerosol in the setting of various air pollutant emissions (hydrocarbons, $NO_x$ and $SO_2$).

In this study, the UNIPAR model was incorporated with the CAMx model (comprehensive air quality model with extensions, v7.10) (Environ, 2020)  to predict SOA formation on regional scales during the Korean-United States Air Quality (KORUS-AQ) campaign that took place between 10 May, 2016 and 10 June, 2016.  During this campaign, inorganic salted aerosols present at four locations across three cities (Seoul, Deajeon, and Gwangju) were wet for the majority of days due to high humidity levels during the night-time and humidity levels above the efflorescent humidity level during the daytime. In

this way, field data accurately portrayed the importance of aqueous phase reactions to predict SOA burdens in ambient air. The organic matter (OM) mass predicted with the CAMx-UNIPAR model was compared with the that predicted with the CAMx model integrated with the conventional two product model (Odum et al., 1996).

## 2 Methods

### 2.1 Integration of UNIPAR with CAMx

The CAMx regional air quality model v7.1  (Environ, 2020) was incorporated with the UNIPAR model as a sub-model to simulate SOA formation during the KORUS-AQ campaign. Fig. 1 illustrates the overall scheme of the integration of the UNIPAR model into the CAMx v7.1 model. For comparison, the CAMx model was simulated with the pre-existing SOAP model by using the two product model (Odum et al., 1996).  The CAMx model internally maps SOA precursors to their gas oxidation mechanism, which uses the State Air Pollution Research Center 07TC (SAPRC07TC) (Hutzell et al., 2012).  For

both the CAMx-UNIPAR and the CAMx-SOAP models used within this study, SOA mass was predicted based on the atmospheric oxidation of aromatics, terpenes, and isoprene. The lumping species in the UNIPAR model were constructed using explicit gas mechanisms that enabled the flexible treatment of multiphase partitioning and aerosol chemistry. Furthermore, the multiphase reaction pathways of organics are important sources of model bias. As a result, the UNIPAR model will improve our ability to accurately estimate SOA mass, which is underpredicted by current regional models. The description of the

UNIPAR SOA module is illustrated in Section 2.2.  For inorganic aerosol species, ISORROPIA calculates the aerosol bulk composition at equilibrium. The SOA mass products sink by dry deposition at the first atmospheric layer to the surface via an irreversible first-order flux.

### 2.2 UNIPAR SOA model

The configuration of the UNIPAR model for SOA formation from each precursor has been described in prior studies

(Beardsley and Jang, 2016; Im et al., 2014; Cao and Jang, 2010; Zhou et al., 2019; Yu et al., 2021b; Han and Jang, 2022). The

predetermined mathematical equations and the model parameters employed in the UNIPAR model have been evaluated in the UF-APHOR chamber for various aromatic volatile organic compounds (VOCs) and biogenic VOCs (terpenes and isoprene) under varying aqueous salted seeds, $NO_x$, $SO_2$, humidity levels, and temperatures. The model equations and parameters used in the UNIPAR model of this study are reported in Section S1 of the Supporting Information. In brief, the key components of the UNIPAR model are described as follows,

1) SOA formation via aqueous phase reactions of organic species is simulated based on the assumption of liquid-liquid phase separation (LLPS) between the organic phase and the salted inorganic solution (Yu et al., 2021a; Im et al., 2014). SOA formation is processed via multiphase partitioning, organic phase oligomerization, and aqueous phase reactions in wet inorganic salted aerosol.

2) The UNIPAR model employs a predetermined mathematical equation to dynamically construct the lumping array, which is linked to the stoichiometric coefficient array of oxygenated products predicted by using near explicit gas mechanisms (MCM v3.3.1 (Jenkin, 2004)) for each precursor. The resulting lumping groups are applied to gas-particle partitioning and heterogeneous reactions in the aerosol phase based on eight volatility ($10^{-8}$, $10^{-6}$, $10^{-5}$, $10^{-4}$, $10^{-3}$, $10^{-2}$, $10^{-1}$, $10^{-0}$ in mmHg) and six reactivity categories defined by their emerging chemistry. These categories are non-reactive (P), slow (S), medium (M), fast (F), very fast (VF), and multifunctional alcohols (MA). Explicit species in UNIPAR lumping include glyoxal (Gly), methylglyoxal (MGly), and isoprene epoxydiols (IEPOX). The UNIPAR model of this study includes 151 lumping species, of which 50 originate from ten aromatics (benzene, toluene, ethylbenzene, propylbenzene, o-xylene, m-xylene, p-xylene, 1,2,3-timethylbenzene, 1,2,4-timethylbenzene, and 1,3,5-timethylbenzene); 50 originate from terpenes; and 51 originate from isoprene.

3) The stoichiometric coefficient array replicates the influence of $NO_x$ on SOA formation by capturing the $RO_2$ chemistry between $RO_2+NO$ reactions and $RO_2+HO_2$ reactions under varying $NO_x$ environments. Additionally, the stoichiometric coefficient array captures dynamically modulated gas under various concentrations of $RO_2$ and $HO_2$ and levels of $NO_x$. The mathematical equations used to construct the stoichiometric coefficient array are reported in Section S1.

4) In the UNIPAR integrated CAMx model, precursor HC consumption is estimated by using the SAPRC07TC gas mechanism (Hutzell et al., 2012). The concentration ($\mu g\ m^{-3}$ of air) of lumping species $i$ is estimated by using the product of the stoichiometric coefficient and each HC consumption value. The resulting concentration is distributed into gas ($C_g$), organic ($C_{or}$), and inorganic phases ($C_{in}$) by using partitioning coefficients and aerosol masses within each phase (Fig. 1). Both the gas-organic partitioning and the gas-aqueous partitioning coefficients are estimated based on Pankow's absorptive partitioning model (Pankow, 1994) with vapor pressure, the estimated activity coefficients of lumping species in each phase of LLPS (Zhou et al., 2019; Jang and Kamens, 1997, 1998), and aerosol's average molecular weight in each phase. The SOA mass formed from the partitioning process (OMP) is attributed to $C_{or}$ and $C_{in}$.

5) The physicochemical parameter arrays, such as molecular weight ($MW_i$), organic to carbon ratio ($O{:}C_i$), and hydrogen bonding ($HB_i$), are used to process multiphase partitioning of lumping species. They are universalized for three major precursor groups (aromatics, terpenes, and isoprene) in the UNIPAR model of this study as described in Section S1.

6)    Both organic phase oligomerization and aqueous reactions of reactive species yield nonvolatile OM in the model. Hence, the heterogeneous reactions in LLPS are operated by two different second order reaction rate constants: $k_{o,i}$ (L mol$^{-1}$ s$^{-1}$) for the organic phase and $k_{AR,i}$ (L mol$^{-1}$ s$^{-1}$) for the aqueous phase. The impact of viscosity on aerosol growth is also considered by including the equation term as a function of the average molecular weight of OM and the O:C ratio (Han and Jang, 2022). Aqueous reactions in the presence of salted solution are operated by acid-catalyzed reactions and OS

formation and are processed under broad ranges of aerosol acidity ([H$^+$]) and relative humidity (RH) levels to form both dry and wet inorganic salted aerosols. In order to simulate SOA mass in ambient air, the chamber generated kinetic parameters are updated by removing the artifact from gas-wall partitioning (Han and Jang, 2020; Han and Jang, 2022).

7)    The SOA mass produced via gas-organic partitioning is estimated using the Newtonian method, which is typically applied in air quality models (i.e., CMAQ) (Schell et al., 2001) based on a mass balance of organic compounds between the gas

and particle phases governed by Raoult's law. Heterogeneously formed nonvolatile OM (OMH) are considered to be a pre-existing absorbing material for gas-particle partitioning (Cao and Jang, 2010; Im et al., 2014). Hence, the gas-organic partitioned equation is modified to include OMH. The SOA mass in the UNIPAR model is attributed to OMP and OMH.

8)    The esterification of sulfuric acid with organic species can form organosulfates (Surratt et al., 2007; Liggio et al., 2005). In particular, dialkylsulfate is non-electrolytic and neutral, and appears in a variety of SOA (toluene, trimethylbenzene,

isoprene, and α-pinene) (Li et al., 2015). In the UNIPAR model, the formation of dialkylsulfate is predicted based on the Hinshelwood-type reaction (Im et al., 2014). The formation of dialkylsulfate reduces aerosol acidity in the presence of sulfuric acid, which catalyzes SOA formation (Jang et al., 2002) and alters aerosol liquid water content (Estillore et al., 2016). In turn, the reduced acidic sulfate due to the dialkylsulfate formation is applied to inorganic thermodynamic model (ISORROPIA) to estimate aerosol acidity and aerosol liquid water content for the next step.

9)    In order to process SOA formation in the inorganic aqueous phase, the inorganic composition and aerosol acidity are predicted by using the inorganic thermodynamic model, ISORROPIA (Fountoukis and Nenes, 2007), and then incorporated into the UNIPAR model. For the ISORROPIA model, mutual deliquescence relative humidity (MDRH) is predicted. In addition, the efflorescence relative humidity (ERH) is predicted using a pre-trained neural network model based on the inorganic composition (Yu et al., 2021a). With MDRH and ERH, the aerosol condition is then estimated to

be wet (organic phase + inorganic aqueous phase) or dry (organic phase + solid-dry inorganic phase). The detailed information for the prediction of aerosol inorganic composition and aerosol acidity are shown in Section S2 in the supporting information.

## 2.3 CAMx configurations

### 2.3.1 Simulation domain and model configurations

The CAMx simulation was conducted using two-way nested grids with a 27-km resolution domain over Eastern Asia (EA) and a 9-km fine resolution domain over South Korea (SK). Fig. 2 displays the simulated domain for this study. For the

vertical domains, 22 vertical layers were simulated for both domains. For comparison, the CAMx model was simulated with two different SOA modules: the original organic gas-aerosol partitioning and oxidation module (SOAP2.2) and the UNIPAR SOA module of this study. The two-mode coarse/fine (CF) scheme for the particle mass distribution was employed. In the CAMx, the multi-section size scheme can be operated with ISORROPIA and SOAP chemistry integrated with CB05 but it is currently not comparable with other gas mechanisms such as SAPRC or other SOA modules (i.e., VBS modules). The dry and wet depositions of aerosols and gas species were estimated with the module existing in the CAMx model for both SOAP and UNIPAR simulations. The detailed explanation for the deposition model can be found in the CAMx User Guide v7.10 (Environ, 2020). All simulations were performed during the time period between 10 May 2016 and 10 June 2016. The detailed configuration of the CAMx simulations is listed in Table 1. The boundary conditions were converted from the MOZART-4 global simulation results (https://www.acom.ucar.edu/wrf-chem/mozart.shtml) (Emmons et al., 2010). The meteorological data used in the CAMx simulation was output from the Weather Research and Forecasting model (WRF). The initial conditions were generated by using a two-week spin-up simulation with the same CAMx model setup. The computation cost for the CAMx-UNIPAR model was evaluated for the one-day simulation of KORUS-AQ data and summarized in Table S1. Overall, the computation time for the CAMx-UNIPAR model is approximately 2 to 2.5 times that of the default CAMx-SOAP simulation depending on the parallel computation setup. The detailed descriptions of statistical parameters to evaluate the model performance are presented in Table S2. Table S3 summarizes the predetermined mathematical equations employed in the UNIPAR to dynamically construct the stoichiometric coefficient array of oxygenated products. The physicochemical parameter arrays for $MW_i$, $HB_i$ and the $O:C_i$ ratios are summarized in Table S4, Table S5, and Table S6, respectively. Table S7 and Table S8 show the kinetic parameters (lumping species' reactivity scales and their basicity constants, respectively) to calculate aerosol phase reaction rate constants in organic phase and inorganic phase.

### 2.3.2 Emission source

The anthropogenic emissions were processed with the Sparse Matrix Operator Kernel Emission (SMOKE) v3.1 (Benjey et al., 2001) for spatiotemporal allocations and chemical speciation as well as vertical allocations for elevated point sources. For areas outside of South Korea, the Northeast Asian emissions inventory contained within the KORUS v5 (Jang et al., 2020b) data was applied. For South Korea, the national emissions inventory named the Clean Air Policy Support System (CAPSS) 2016 (http://airemiss.nier.go.kr/mbshome/mbs/airemiss/index.do) was applied (Choi et al., 2020; Lee et al., 2011). The Model of Emissions of Gases and Aerosols from Nature (MEGAN) v2.04 (Guenther et al., 2006) was used to prepare the biogenic emissions. Since the chemical speciation profiles for the emissions inventory were only available for the SAPRC99 chemical mechanism, some of the model VOC species were mapped into those in the SAPRC07TC, with which the CAMx model was operated in this study.

**2.4 Observations during the KORUS-AQ campaign**

The KORUS-AQ field study was conducted by the joint efforts of the National Institute of Environmental Research of South Korea and the National Aeronautics and Space Administration (NASA) of the United States to understand the factors controlling air quality across urban, rural, and coastal interfaces. The study integrated observations from aircraft, ground sites, and satellites with air quality models. The ground-level observational data used in this study were performed at four different monitoring stations located in three cities within South Korea (Crawford et al., 2021). Table 2 lists the geological locations of the monitoring sites, the sampling times, and the measured chemical species used in this study. The Olympic Park site is a green site for recreational and sports activities that is heavily influenced by pollution from surrounding urban traffic and buildings in Seoul. The Bulkwang site is a highly populated residential area located in the far north-western corner of Seoul. The Daejeon site is a populated downtown location within a metropolitan city. The Gwangju site is a residential and forest area located far north of downtown Gwangju.

The organic carbon concentration (OC) was continuously monitored by using the semi-continuous organic carbon/elemental carbon analyzer (OCEC, Sunset Lab. Inc). An averaged OM/OC factor of 1.5 was applied to estimate the OM concentration at the observation sites (Park et al., 2018). The concentrations of water-soluble inorganic ions were monitored by using an ambient ion monitor (AIM) at the Bulkwang supersite, the Daejeon Supersite, and the Gwangju Supersite. The Monitor for AeRosols and Gases in Ambient air (MARGA ADI 2080, Metrohm, Switzerland) was used to measure the inorganic compositions at the Olympic Park supersite. $O_3$ and $NO_x$ levels were monitored by using an $O_3$ analyzer (EC9810, Ecotech, Australia) and a $NO_x$ analyzer (EC9841, Ecotech, Australia), respectively. VOCs were monitored by using a Gas Chromatography-Flame Ionization Detector (Varian GC450). The hourly averaged observations and simulations of sulfate, nitrate and ammonium concentrations are displayed in Fig. S1-S3. The eight-hour averaged concentrations of $O_3$, $NO_x$, and $SO_2$ are displayed in Fig. S4. In Fig. S5, the simulated concentration of SOA precursors, including toluene, benzene, and isoprene, are plotted against the observations at the Olympic Park supersite. For isoprene, the observation was not available. For meteorological inputs, the observed temperature and RH at the Olympic Park supersite are plotted versus the simulations in Fig. S8. Overall, the smaller bias between observations and predictions appeared in temperature compared to RH.

The KORUS-AQ campaign performed several flight measurements by using the NASA DC-8 research aircraft with a comprehensive payload for in situ sampling of trace gas and aerosol compositions. Fig. S11 shows the flight tracks of the NASA DC-8 aircraft missions during the KORUS-AQ campaign between May 10 and June 10 in 2016. The observed airborne concentrations of $O_3$, $NO$, $NO_2$, and toluene are plotted against the simulation from the CAMx-UNIPAR model (Fig. S12).

## 3 Results and discussion

### 3.1 Simulated concentration of OM

The OM concentrations, which were simulated by using the CAMx model with two different SOA module setups (SOAP and UNIPAR), were plotted with the ground-based observation data at the four different monitoring sites (the Bulkwang Supersite, the Olympic Park Supersite, the Daejeon Supersite, and the Gwangju Supersite) between 10 May 2016 and 10 June 2016 during the KORUS-AQ campaign (Fig. 3). Table 2 summarizes the organic products used to estimate the SOA formation in both the SOAP and UNIPAR models. For the SOAP simulation, the OM is the sum of the primary organic matter (POM) and SOAP products (SOA1, SOA2, SOA3, SOA4, SOPA, and SOPB). In the UNIPAR module, OM is predicted as the sum of POM and SOA produced from 151 lumping species. The POM is a single non-volatile species that is not involved in the aerosol phase chemistry. POM and non-volatile SOA mass can both influence the gas-particle partitioning process.

For all four sites, the ground-based OM concentrations were noticeably high between 24 - 31 May 2016. The CAMx model equipped with both SOAP and UNIPAR mechanisms generally simulate this tendency. Fig. 4 illustrates the hourly averaged OM concentrations during this high OM period. A sharp increase of OM appears during the daytime in all field data. This tendency is captured by the CAMx model in both the Olympic Park (Fig. 4b) and Daejeon sites (Fig. 4c) but is not well predicted at the other two sites (Fig. 4a and Fig. 4d). The rapid increase in OM concentrations during the daytime can be explained by the diurnal pattern of pollutant emission due to human activities and photochemically produced SOA. Overall, the CAMx-UNIPAR simulation better predicts field observations than the CAMx model with the SOAP module, although both the UNIPAR and SOAP models underestimate high OM data points. The underestimation of high OM peaks is potentially due to the missing precursors in the emission inventories. For example, the CAMx simulation used in this study was performed with aromatics, terpenes, and isoprene (Table 2). The SOA simulation needs to be updated to include sesquiterpenes, intermediate VOCs, and volatile chemical species sourced from residential, commercial, and industrial sectors (Mcdonald et al., 2018).

Fig. 5 displays the scatter plot for the hourly averaged OM observations vs. predictions at four different ground-based monitoring sites. The statistical parameters were calculated for each site by using the mean bias error (MBE), the Pearson correlation coefficient (PCC), the mean fractional bias (MFB), and the mean fractional error (MFE). The detailed descriptions of these statistical parameters are presented in Table S2. For the Bulkwang, Daejeon, and Gwangju supersites, the estimated MBE value is between -1.59 to -0.50 indicating that the OM concentration is slightly underestimated in the simulation. The absolute values of MBE, which indicates proximity to observations, are smaller in the CAMx-UNIPAR model for Bulkwang (-0.80), Daejeon (-0.50), and Gwangju (-1.02) than those in the CAMx-SOAP models for Bulkwang (-1.43), Daejeon (-1.01), and Gwangju (-1.59). In general, the model performance were proposed as MFB $\leq \pm$ 30% with MFE $\leq \pm$ 50% for the best model performance; and MFB $\leq \pm$ 60% with MFE $\leq \pm$ 75% for the acceptable model performance (Boylan and Russell, 2006). For all the simulations performed with the CAMx-UNIPAR model, the simulation results meet the best model performance goal. Additionally, the estimated MFBs with the CAMx-UNIPAR model at Bulkwang, Daejeon and Gwangju supersites are

lower than those with the CAMx-SOAP model. The estimated MFE values with the CAMx-UNIPAR simulation are also 3-10% lower than those with the CAMx-SOAP model. For the Olympic Park supersite, the simulations by both the SOAP and UNIPAR modules yield positive MBE (0.3 and 0.89, respectively) and MFB (17.7% and 34.4%, respectively) values indicating that OM is overestimated. This overestimation can be associated with the variabilities in local emissions at the Olympic Park, which is a green space in an urban area. Thus, the ground pollution levels measured at this site may be lower than those observed in the city center. However, the model simulation is mainly influenced by the polluted urban air within a 9-km fine resolution domain. For the PCC values, no significant differences are observed between the values generated by the SOAP and UNIPAR simulations. For organic matter, the average Normalized Mean Bias (NMB, %) between model predictions and observations at the four monitoring sites are -50% for CAMx-SOAP and -39% for CMAx-UNIPAR. A similar level of the NMB ($\approx$ 46%) was reported in the previous simulation for the same campaign (Park et al., 2021).

### 3.2 Impact of aqueous reactions on SOA mass

Aqueous reactions of reactive organic species in the presence of salted wet aerosol play an important role in the formation of SOA (Volkamer et al., 2007; Volkamer et al., 2009). Therefore, the quantity of aerosol liquid water content is an important model parameter to simulate aqueous reactions. In the CAMx-UNIPAR modules, the phase state of inorganic aerosol is determined by using the calculated ERH based on the pre-trained neural network mathematical equation (Yu et al., 2021a) and the inorganic compositions (sulfate, ammonium, nitrate, and protons) predicted from the inorganic thermodynamic model, ISORROPIA (Fountoukis and Nenes, 2007). When inorganic aerosol is wet, the aerosol liquid water content is calculated by using the inorganic thermodynamic model.

As described in Section 2.2, dialkylsulfate formed in the presence of acidic sulfate is neutral and subtracted from sulfates in the inorganic thermodynamic model. However, during the KORUS-AQ campaign periods, ammonia was abundant at the four observation sites, which neutralized acidic aerosol. The low aerosol acidity during the KORUS-AQ campaign has been reported in prior studies (Yu et al., 2020; Nault et al., 2021). Under these circumstances, neither SOA formation via acid catalyzed aqueous reactions nor dialkylsulfate formation is effective. Evidently, the predicted sulfate concentrations with the CAMx-UNIPAR model are comparable to those with the CAMx-SOAP model.

As seen in Fig. 5, temperature at night-time considerably decreased and this influenced the diurnal pattern of RH. The time profile of the predicted phase state of inorganic aerosols at the four observation sites is shown in Fig. S7. When RH is higher than DRH of inorganic salts, the aerosol is wet and can process heterogeneous reactions of organics in the aqueous phase (Witkowski et al., 2018). The inorganic aerosols were wet for the observation periods for 10 -18 May, 23 -31 May, and 7 -10 June. The time profiles of POM, OMH, and the OM produced by multiphase partitioning (OMP) are shown in Fig. 6. The concentration of POM is insensitive to RH while OMH and OMP are strongly impacted by the inorganic aerosol phase state. For example, OMH and OMP are significantly increased during the wet period (24-28 May) compared to those in the dry period (18-22 May). During the wet period, the concentration of anthropogenic HC increased. However, the increased OM during the wet period cannot be explained solely by the increased HC emissions. When the mass concentrations of both wet

inorganic salts and preexisting OM (POM+OMH) are high, OMP increases. In addition, the observed RH values at the Daejeon (Fig. S7c) and Gwangju sites (Fig. S7d) are generally higher than those at the Bulkwang (Fig. S7a) and Olympic Park sites (Fig. S7b). Consequently, the simulated OMH fraction to the total OM is greater at the Daejeon and Gwangju sites than those at the two other sites in Seoul (Fig. 6).

290        The observed OM concentrations at Bulkwang and Olympic Park sites was plotted vs. the predicted OM concentrations for the wet period (Fig. S9a and S9c) and dry period (Fig. S9b and S9d). The date and the duration for dry and wet periods can be found in Fig. S7. Overall, the estimated MFB and MFE values were similar between wet and dry aerosols. All simulations met the best model performance goal as described in Section 3.1 (MFB $\leq \pm 30\%$ with MFE $\leq \pm 50\%$). For the Bulkwang site, the deviation of predicted OM from observations was less with wet aerosol than with dry aerosol, yielding a smaller MBE and a greater PCC in wet periods than in dry periods. The POM fraction to total OM is high during dry periods and thus, the contribution of POM uncertainties to total OM can be high leading to the small PCC (0.2). During wet time, the simulated SOA fraction to total OM is large due to salted aqueous phase and this can reduce the influence of POM uncertainties on total OM (PCC = 0.66). This difference in PCC values between the dry period and the wet period at the Bulkwang site clearly shows the essential role of aqueous reactions on OM prediction. For the Olympic Park site, no significant difference in PCC was calculated between the wet and dry periods within 5%.

**3.3 SOA compositions and spatial distribution**

       Fig. S14 displays the spatial distribution of the averaged OM concentrations (Fig. S14a) for each SOA species (Fig. S14b-S14h) for one month (10 May to 10 June 2016) during the KORUS campaign period. The averaged OM ranged from 0 to 10 $\mu g$ m$^{-3}$ (Fig. S14a). Notably, the relatively high OM concentrations in the range of 4-10 $\mu g$ m$^{-3}$ was observed in Southeastern China as well as the China's coastal regions along the Yellow Sea. The high OM concentrations in Southeastern China are associated with the large quantity of biogenic VOCs emitted in the region (Fig. S14e and S14f) in the presence of anthropogenic NO$_x$. POM is high near China's East coastal zone because anthropogenic emissions are high from industrial areas of Shandong Province. Evidently, the contribution of aromatic SOA (Fig. S14c and S14d) is also high near the Yellow Sea due to long-range transport of anthropogenic pollution. Of aromatic SOA, the OMP fraction (Fig. S14d) is significantly greater than OMH (Fig. S14c) due to the hygroscopic sea spray aerosol under high humidity levels over the Yellow Sea. The overall isoprene SOA contribution is relatively small within the simulated domain (< 0.5 $\mu g$ m$^{-3}$) (Fig. S14g and S14h).

       The observations gathered at the four sampling sites of this study can be impacted by both local and long-range transport from China's industrial areas. In particular, light southwesterly winds were found within the boundary layer behind the front, facilitating steady transport of air pollution from China into the study regions during the low-level transport/haze period (24-31 May 2016) (Crawford et al., 2021). The previous analysis of data collected by a high-resolution time-of-flight aerosol mass spectrometer (HR-ToF-AMS) (Kim et al., 2018) showed that the stagnant period (17-22 May 2016) was found to be driven by OM, whereas particulate matter during the low-level transport/haze period was dominated by the inorganic aerosol components due to a combination of transport and the metrological condition to form haze. The CAMx-UNIPAR

simulation results are supported by this field analysis, revealing the increased SOA mass in the presence of the high
concentration of wet inorganic aerosol during the low-level transport/haze period. The SOA precursors in this study include
aromatics, terpenes, and isoprene as shown in Table 2. Fig. 7 illustrates the time profile of the hourly averaged concentrations
of SOA species for the four observation sites. In the CAMx-UNIPAR prediction, the high POM (Fig. 6) during the period of
the low-range transport/haze development increases OMP and sequentially elevates OMH. In the similar manner, the high
POM can increase SOA mass with the CAMx-SOAP model. Additionally, the aromatic SOA fractions, predicted by the
UNIPAR model, were high during this time period across all four sites. The contribution of biogenic SOA to local OM burden
varies by location. For example, the biogenic SOA mass fractions of total SOA at the Gwangju site are obviously higher than
those present at the other three sites because the Gwangju site is in a suburban environment surrounded by farming and forested
areas.

In Fig. 7a-7d (UNIPAR), OMH attributes to 22% to 48% of aromatic SOA, showing the importance of heterogeneous
reactions of aromatic products to form SOA during the KORUS-AQ campaign. A large fraction of OMP predicted by the
CAMx-UNIPAR model during the wet period is attributed to the partitioning mass to salted inorganic aqueous phase. Based
on the internal calculation of the UNIPAR simulation, the OMP associated with the aqueous phase comprises higher than 80%
of the total OMP. In general, aromatic SOA is more sensitive to aqueous reactions than terpene SOA. Particularly, hydrophilic
aromatic products (O:C ratios: 0.65-0.95) greatly contribute to OMP during the haze period (24-31 May). Terpene-derived
SOA is relatively less polar (O:C ratios: 0.45-0.65) and less volatile than oxygenated aromatic products. Therefore, terpene
SOA is minimally impacted by the aqueous phase due to its poor solubility in salted aqueous solution. The OMP fraction of
terpene SOA is significantly less compared to that in aromatic SOA. Isoprene SOA is negligible at all sites due to low isoprene
emissions. An estimation of biogenic hydrocarbon emissions in the global scale, simulated by Sindelarova et al. (2014) by
using Megan for 30 years, showed that the relative significance of isoprene emission is little in South Korea. Isoprene SOA is
known to be highly sensitive to aerosol acidity that can accelerate SOA formation (Pye et al., 2013; Lewandowski et al., 2015;
Beardsley and Jang, 2016). During the KORUS-AQ campaign (Section 3.2) period of this study, inorganic acids were nearly
neutralized and thus, the effect of acid catalysed reactions on SOA may be trivial. The SOA mass simulated by SOAP (Fig.
7e-7h) is mainly attributed to partitioning mass originating from SOA1, SOA2, SOA3 and SOA4. The contribution of
nonvolatile SOA mass (SOPA+SOPB) to total SOA mass in the SOAP model is small compared to that (OMH to SOA mass)
predicted by the UNIPAR model.

### 3.4 Simulated concentrations of gaseous species

Fig. S10 illustrates the correlation between the 8-hour averaged observations and the 8-hour averaged predictions of
$O_3$, $NO_x$, $SO_2$ and toluene at the Olympic Park supersite. In general, the model prediction slightly underestimates $O_3$ (Fig.
S10a), $SO_2$ (Fig. S10c), and toluene (Fig. S10d), but overestimates $NO_x$ (Fig. S10b). Similarly, underestimation of $O_3$ appeared
in the on-board data (Fig. S12a). This underestimation could be explained by the missing or the underestimation of $O_3$
precursors (i.e., toluene as shown in Fig. S12d) in the current emission inventories.

Fig. S13 shows the correlation between Aerosol Mass Spectrometer (AMS) data and the simulated primary organic aerosol (POA) or the simulated secondary organic aerosol (SOA). A higher correlation coefficient appears between the AMS data collected during the DC-8 flight missions and the simulated SOA (PCC = 0.57) than that between AMS data and the simulated POA (PCC = 0.38), indicating that observed OC in the high altitude is more influenced by secondary pollutants.

## 4 Atmospheric implications and uncertainties

There is no feature in the two-product model (SOAP) to process aqueous reactions in the presence of salted aqueous phase. Thus, the CAMx-SOAP simulation may lead to OM underestimation. For example, OM was underpredicted in Europe during winter seasons, which caused frequent fog developments to form wet aerosol (Meroni et al., 2017; Jiang et al., 2021). SOA formation has also been underestimated in the Southeast United States where isoprene emissions are high during wet summer seasons (Marais et al., 2016). In particular, isoprene SOA formation is known to be largely influenced by aerosol acidity and aerosol water content (Pye et al., 2013; Lewandowski et al., 2015; Beardsley and Jang, 2016). In this study, the amount of OM was simulated by using both the newly derived CAMx-UNIPAR and preexisting CAMx-SOAP models. The simulated results were compared to the ground-based observations at four observation sites during the KORUS-AQ campaign (Fig. 3). Under the dry period (Fig. 3), the predicted SOA mass by the UNIPAR model is dominated by gas-particle partitioning onto organic phase and oligomerization in organic aerosol. During the wet period, SOA production forms mainly through gas-aqueous partitioning and aqueous reactions. The KORUS-AQ campaign took place during the wet period of the typical summer climate in South Korea. Thus, SOA formation can be considerably promoted by aqueous reactions as discussed in Section 3.2.

Overall, the statistical parameters (Fig. 5) to predict OM showed a better model performance with the CAMx-UNIPAR model compared to the CAMx-SOAP model. The predicted POM concentrations were relatively steady during the campaign period while the simulated SOA concentrations varied with environmental conditions and emission profiles (Fig. 6). The SOA mass predicted by the CAMx-UNIPAR model was sensitive to humidity levels and emission profiles and contributed to the accurate prediction of OM (Fig. 3 and Fig. 4). For example, the contribution of aromatic SOA to total OM increased (Fig. 7) during haze episodes (24-31 May 2016) owing to aqueous reactions and partitioning of polar aromatic products into salted aqueous solution.

The major SOA precursors included in the CAMx-UNIPAR model of this study were aromatics, terpenes, and isoprene. These precursors are major contributors of urban SOA but other precursors such as sesquiterpene, alkanes, and polyaromatic hydrocarbons are missing in the current simulation of CAMx-UNIPAR. For example, long-chain alkanes or intermediate VOCs are often found in the urban air associated with incomplete combustion and diesel exhaust (Worton et al., 2014; Perrone et al., 2014; Alam et al., 2018). In the future, the UNIPAR model would benefit from an update to include the model parameters of missing precursors. The inclusion of missing precursors into the CAMx-UNIPAR model can address the underestimation of OM as compared to field observations. In addition, the recent study by Mcdonald et al. (2018) showed that volatile chemical products (>53% of total anthropogenic VOC emissions in Los Angeles, USA) originating from consumer

and industrial products, which are currently unaccounted for in models, can significantly contribute to SOA burden in the urban atmosphere. In addition, the deposition of SOA was estimated with the one particle size bin. The different particle size can have different sink fluxes causing the uncertainty in the lifetime of OM. The UNIPAR model is capable of predicting aging of gas products but currently has no feature for OM aging.

During the KORUS-AQ campaign period, particle-phase inorganic nitrate was high when RH was high, or temperatures dropped in the afternoon (Fig. S6 and S7). Synergistically, high particulate nitrate can modify aerosol hygroscopicity and increases aerosol water content, consequently elevating SOA formation through aqueous reactions. The current UNIPAR model is capable of processing multiphase partitioning of organic species and determining phase state in the presence of ammonium-sulfate-nitrate aerosol. The degree of gas aging is also encompassed in UNIPAR model as a function of the quantity of atmospheric radicals ($HO_2$ and $RO_2$), which are simulated in the gas mechanism. However, the prediction of these radicals varies with different gas mechanisms (i.e., Carbon Bond mechanisms (Tanaka et al., 2003; Yarwood et al., 2005; Yarwood et al., 2010), SAPRC mechanisms (Carter, 2010), and Master Chemical Mechanism (Jenkin et al., 2012; Jenkin, 2004)) and thus, variably influences SOA prediction. The accuracy of the prediction of aerosol acidity can also affect the prediction of SOA formation, because of acid-catalysed oligomerization of organic species. In general, aerosol acidity tends to be under-predicted in ammonia-rich aerosol (Li et al., 2015; Jang et al., 2020a). In the future, the performance of the CAMx-UNIPAR model needs to be evaluated by simulating various episodes compared to field studies.

*Code availability.* Code to run the CAMx-UNIPAR model in this study is available upon request.

*Data availability.* Regional simulation input data is available upon request.

*Author contributions.* ZY simulated the air quality during the KORUS-AQ campaign by using the CAMx-UNIPAR model. MJ structured the UNIPAR model and integrated UNIPAR to CAMx. SH and AM provided model parameters for UNIPAR. SK, KS, and JP provided meteorology and emission inputs to CAMx model simulations.

*Competing interest.* The authors declare that they have no conflict of interest.

**Acknowledgments**

This research was supported by the National Institute of Environmental Research (NIER2020-01-01-010); the National Science Foundation (AGS1923651); and the Fine Particle Research Initiative in East Asia Considering National Differences (FRIEND) Project through the National Research Foundation of Korea (NRF) funded by the Ministry of Science and ICT (2020M3G1A1114556). We also thank Dr. Ross Beardsley from Ramboll USA, Inc. (CA, USA) for the supporting the setup of CAMx simulation.

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

**Table 1.** List of KORUS-AQ campaign observations sites.

| Site Name | Location (Latitude/Longitude) | Time period | Measurements [a] |
|---|---|---|---|
| Bulkwang Supersite | 37.61/126.93 | 8 May – 12 June | OC, inorganic ions |
| Olympic Supersite | 37.52/127.12 | 9 May – 13 June | OC, inorganic ions, $O_3$, $NO_x$, $SO_2$ |
| Daejeon Supersite | 36.35/127.38 | 8 May – 12 June | OC, inorganic ions |
| Gwangju Supersite | 35.23/126.84 | 8 May – 12 June | OC, inorganic ions |

[a] The detailed instruments used for the observations are described by Crawford et al. (2021).

**Table 2**. List of SOA products in the SOAP and UNIPAR modules.

| Mechanism | Species | Number of species | Note |
|---|---|---|---|
| SOAP | SOA1 | 1 | High volatility aromatic SOA products |
| | SOA2 | 1 | Low volatility aromatic SOA products |
| | SOPA | 1 | Non-volatile aromatic SOA products |
| | SOA3 | 1 | High volatility biogenic SOA products |
| | SOA4 | 1 | Low volatility biogenic SOA products |
| | SOPB | 1 | Non-volatile biogenic SOA products |
| UNIPAR | OMH-ar | 50 | Heterogeneously formed aromatic SOA |
| | OMP-ar | 50 | Partitioned aromatic SOA |
| | OMH-te | 50 | Heterogeneously formed terpene SOA |
| | OMP-te | 50 | Partitioned terpene SOA |
| | OMH-is | 51 | Heterogeneously formed isoprene SOA |
| | OMP-is | 51 | Partitioned isoprene SOA |

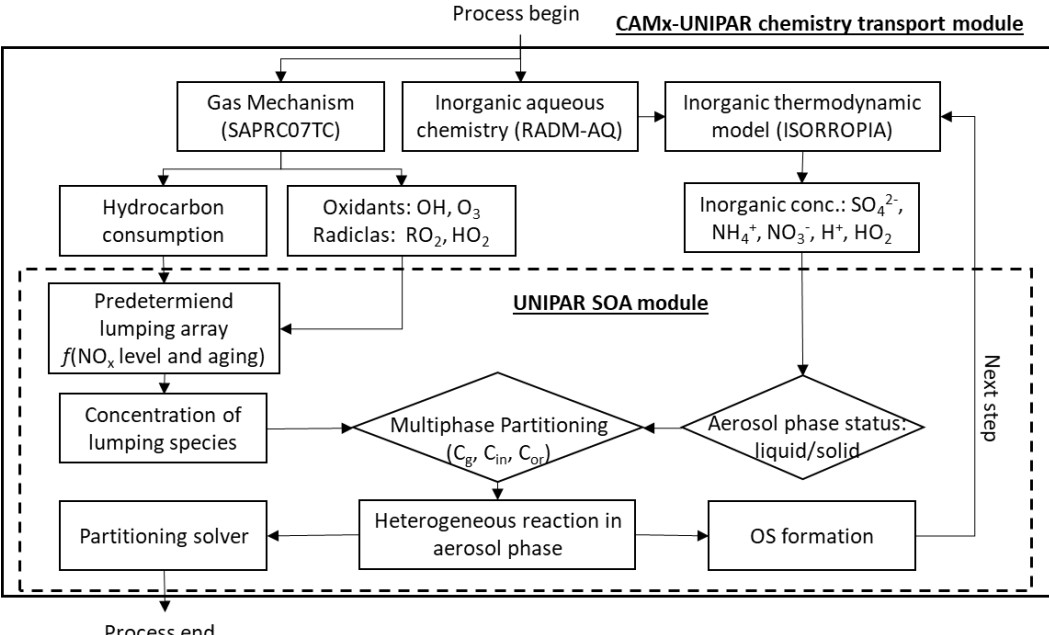

**Figure 1:** Scheme of CAMx v7.1 chemistry transport model integrated with the UNIPAR secondary organic aerosol module.

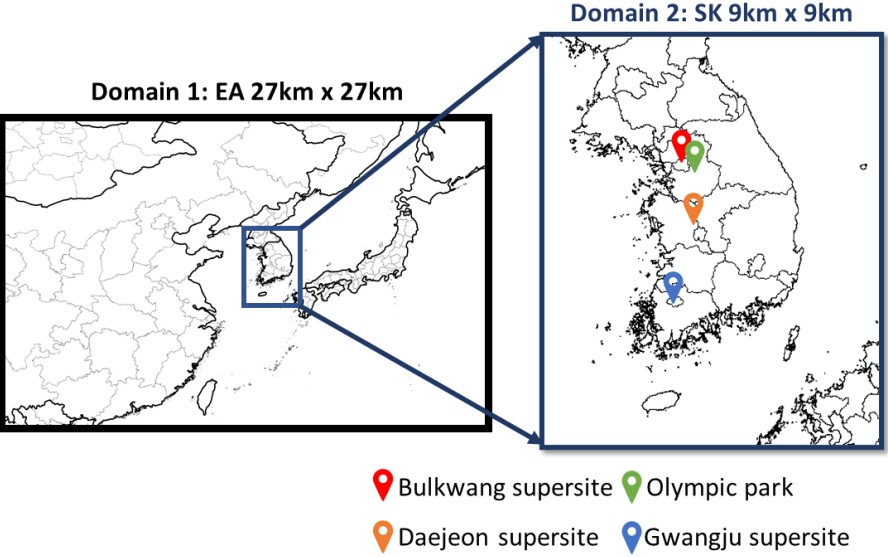

**Figure 2:** Simulated domain and observation sites during the KORUS-AQ campaign. "EA" and "SK" represent Eastern Asia and South Korea, respectively.

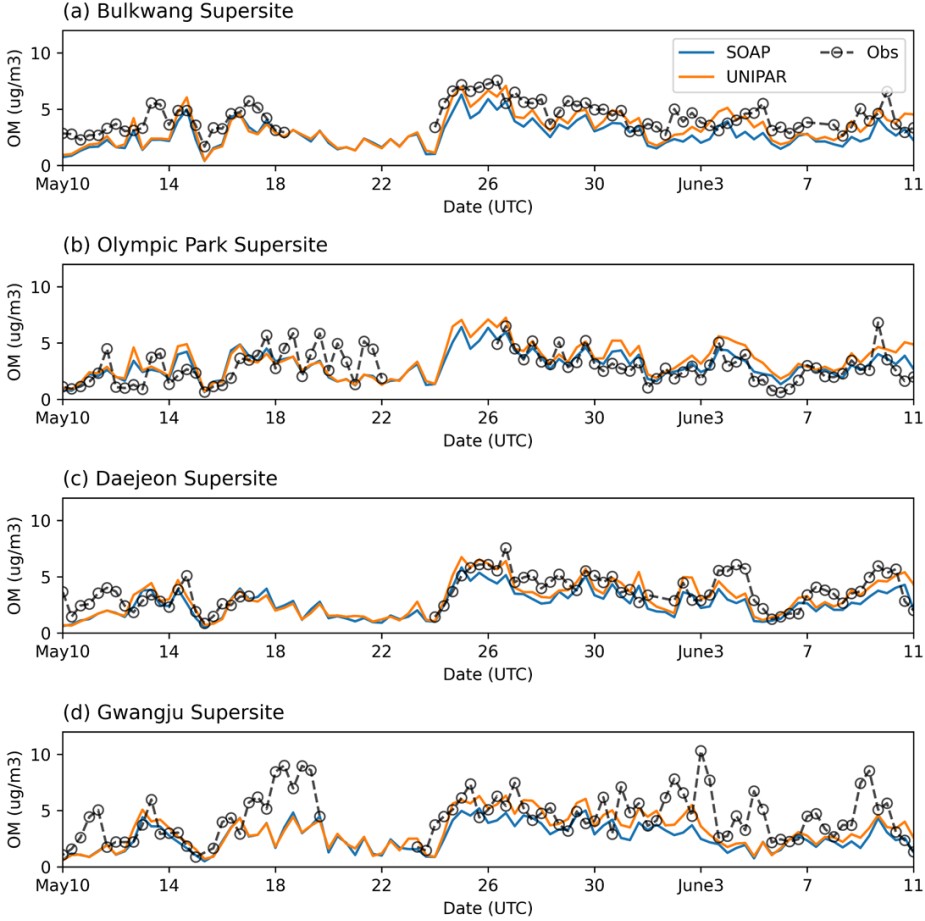

**Figure 3:** Time profiles of OM concentration (µg m$^{-3}$) averaged over eight hours for the observation data and the CAMx simulation results at the (a) Bulkwang, (b) Olympic Park, (c) Daejeon, and (d) Gwangju supersites.

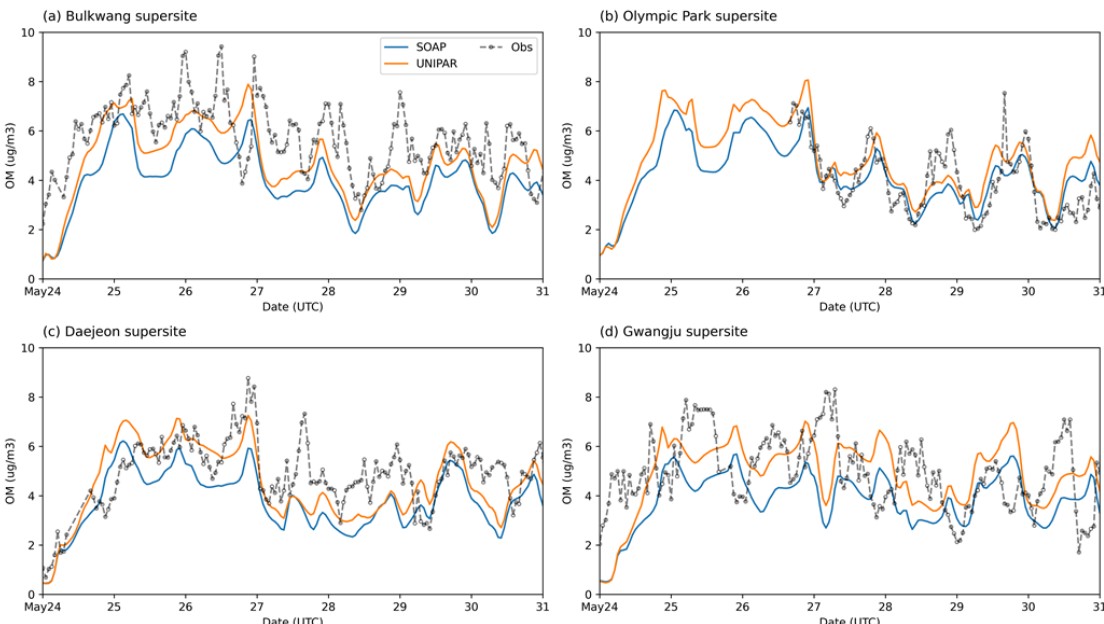

**Figure 4:** Time profiles of hourly averaged OM concentrations ($\mu$g m$^{-3}$) for the observation data and the CAMx simulation results at the (a) Bulkwang, (b) Olympic Park, (c) Daejeon, and (d) Gwangju supersites.

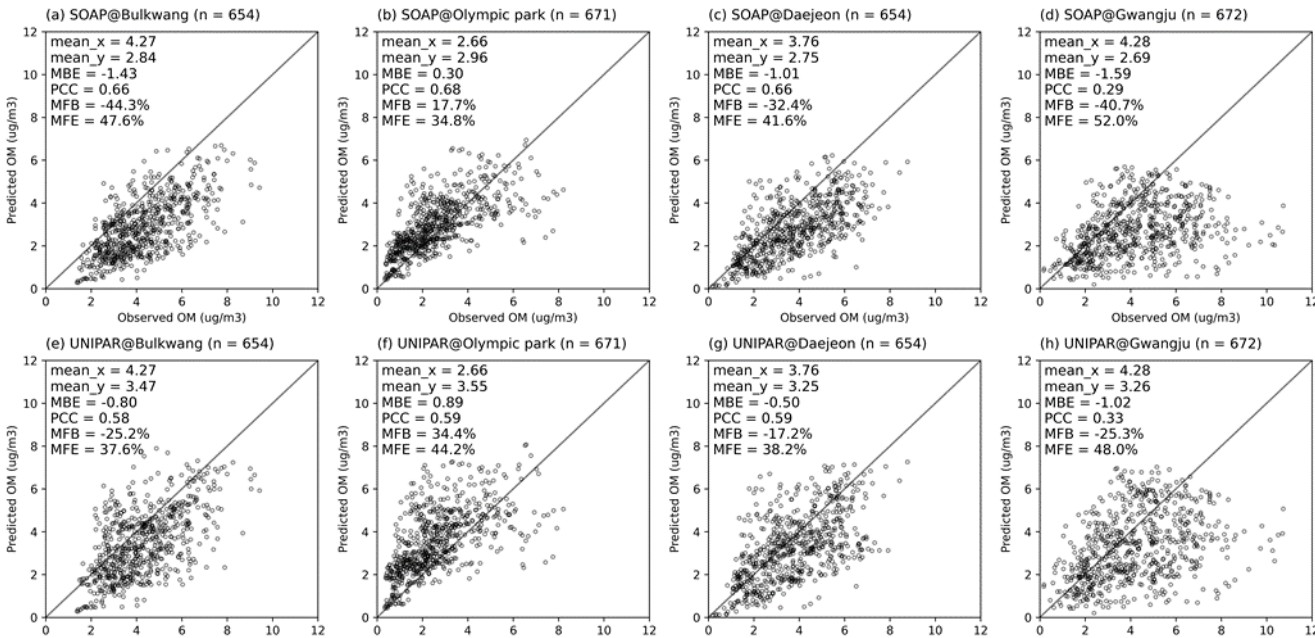

**Figure 5.** The observation of hourly averaged OM concentrations vs. CAMx simulation results by using the SOAP SOA module at the (a) Bulkwang, (b) Olympic Park, (c) Daejeon, and (d) Gwangju supersites; and the results by using the UNIPAR SOA module at the (e) Bulkwang, (f) Olympic Park, (g) Daejeon, and (h) Gwangju supersites. "mean_x" and "mean_y" are the averaged OM concentrations of observations and predictions, respectively. "MBE" is the estimated mean bias error. "PCC" is the Pearson correlation coefficient. "MFB" and "MFE" are the mean fractional bias and mean fractional error, respectively.

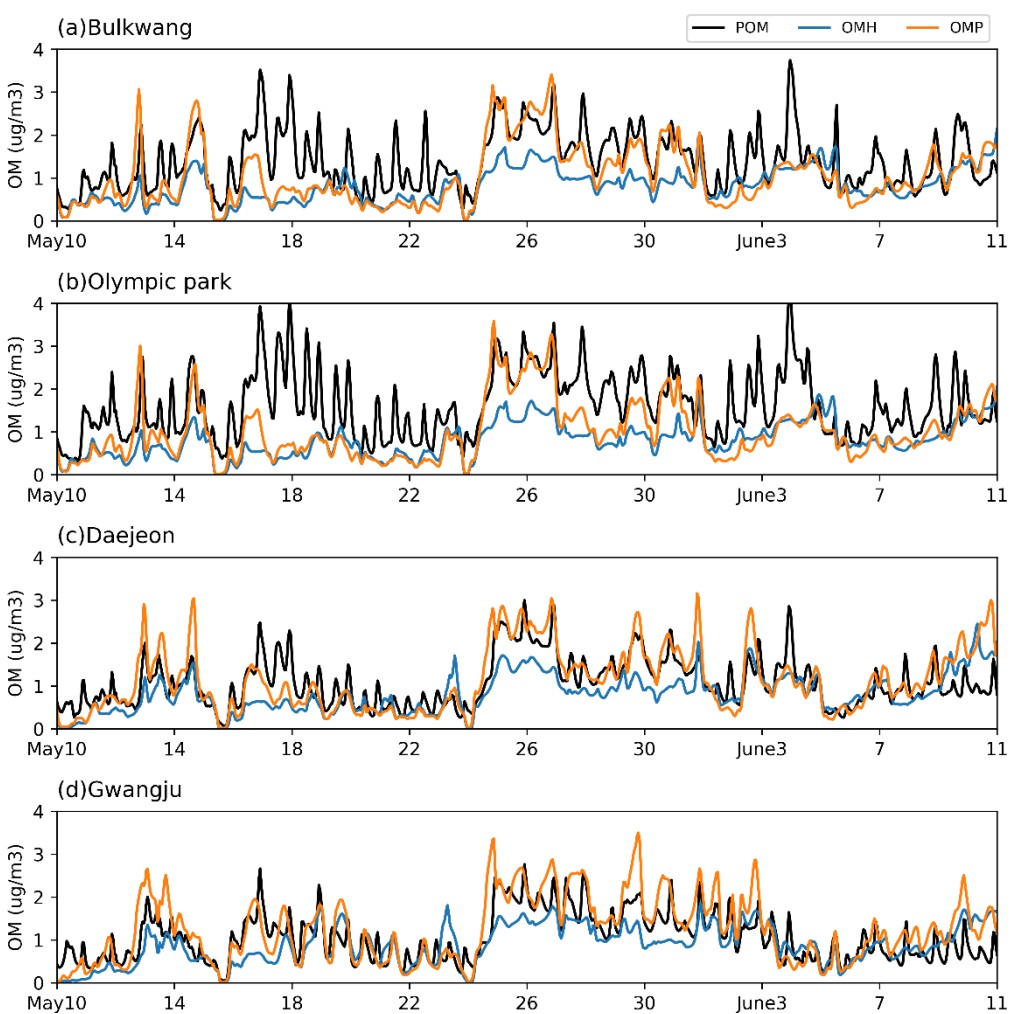

**Figure 6.** Time profiles of the predicted hourly averaged concentrations of primary organic matter (POM), OM formed via heterogeneous reactions (OMH), and OM formed via gas-particle partitioning process at the (a) Bulkwang, (b) Olympic Park, (c) Daejeon, and (d) Gwangju supersites.

655

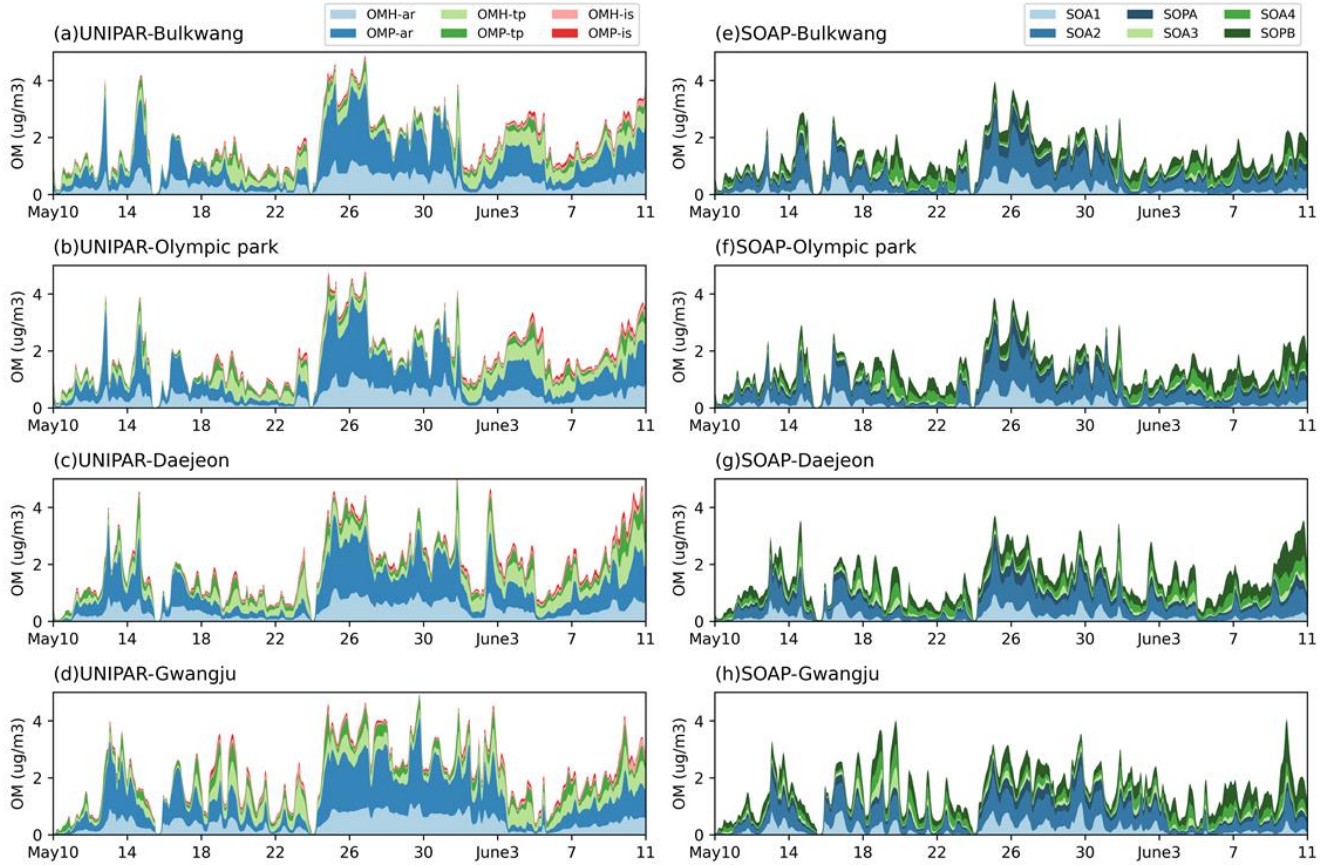

**Figure 7.** Time profiles of stacked OM concentrations predicted by the CAMx-UNIPAR model at the (a) Bulkwang, (b) Olympic Park, (c) Daejeon, and (d) Gwangju supersites; and by the CAMx-SOAP model at the (e) Bulkwang, (f) Olympic Park, (g) Daejeon, and (h) Gwangju supersites. "OMH" and "OMP" represent the OM formed through heterogeneous reactions and gas-particle partitioning process, respectively. "ar", "te", and "is" represent the SOA formed from aromatic hydrocarbons, terpenes, and isoprene, respectively.