# Peer review of "Secondary Organic Aerosol Formation via Multiphase Reaction of Hydrocarbons in Urban Atmospheres Using the CAMx Model Integrated with the UNIPAR model"

_Atmospheric Chemistry and Physics, 2021_

## Author Comment (AC1)

**Manuscript #: acp-2021-1002**

**Response to Anonymous Referee #RC1:**

In this manuscript, the authors incorporated their sophisticated SOA model (UNIPAR) with an air quality model (CAMx) and simulated SOA concentrations from different formation pathways and different precursors. Observed concentration of organic matter (OM) is better reproduced by the UNIPAR mode than by a conventional two product model (SOAP). By applying the UNIPAR model, the SOA formation from gas-particle partitioning, inparticle oligomerization, and aqueous-phase reactions are separately calculated, and their contributions have been quantified.

This manuscript is well written and includes useful information about the numerical modeling of SOA formation processes in the ambient air. However, I have several concerns as below. I recommend this manuscript for publication after the following concerns are adequately addressed.

Response: We thank the reviewer for the valuable comments on this manuscript. To response to the comments from the reviewer, the explanation and discussions are added in the revised manuscript. A line-by-line response for each comment are listed below.

**1 Comments on Methodology**

I am afraid that methodology (model and emissions) is not comprehensively described, or adequate references are cited.

• You wrote in L109 that "The mathematical equations used to construct the stoichiometric coefficient array are reported in Section S1" and four parameters (A, B, C, and D) for different precursors and conditions (NOx level and aging status) are given in Table 3. However, I could not find the information how did you consider dependence on NOx (high/low) and aging degree (fresh/aged) for the calculation of stoichiometric coefficients in the ambient conditions.

Response: The information for the impact of the  $NO_x$  level and aerosol aging has been added to the section S1 in the supporting information and reads now.

"The mass-based stoichiometric coefficient ( $\alpha$ i) of each lumping species i can be calculated based the variables listed in Table S3. Both the stoichiometric coefficient array derived from the fresh compositions and that from the aged compositions are determined as a function of NOx levels using the mathematical equations. To simulate age-dependent SOA formation, the stoichiometric coefficient array is reconstructed over time by a weighted average of fresh and aged stoichiometric coefficients based on the normalized concentration of oxidized organic radicals and HO2 with a hydrocarbon concentration. The detailed information of the calculation of age-dependent stoichiometric coefficient of lumping species was discussed in the previous study (Zhou et al., 2019)." • You set six categories for oxidation products: non-reactive (P), slow (S), medium (M), fast (F), very fast (VF), and multifunctional alcohols (MA). Products with these categories are always produced or did you consider any condition dependence?

Response: The value of the stoichiometric coefficient associated with volatility and reactivity in aerosol phase depends on the precursors, the oxidation status (aging), and the  $NO_x$  levels. It is not necessary that all stoichiometric coefficient arrays are filled. For example, the quantity of MA is high in isoprene products but very little or none in the products from other precursors.

• Thermodynamic parameters of oxidation products (vapor pressure and vaporization enthalpy) are not explicitly shown.

Response: Prior to the establishment of the physicochemical parameters (vapor pressure, enthalpy of vaporization) of the UNIPAR lumping species, the physicochemical parameters of all explicit products were individually calculated and classified into vapor pressure groups which is paired with enthalpy of the vaporization. In UNIPAR, the stoichiometric coefficients associated with volatility was not semi-empirically determined but determined by considering the properties of explicit products.

• Information of emission amounts is not shown. As you estimated the contributions of SOA precursors, total emissions or their distributions are important information. I have two more concerns about emissions:

Response: As seen in Section 2.3.2, the emission of air pollutants was prepared by using SMOKE from emission inventories originating from various sources (i.e., point sources, area sources, biogenic sources (MEGAN), automobiles non-mobile sources, etc.).

- You wrote in L255 that "During the wet period, HC emissions increased". It appears from Figures S5 and S6 that daytime temperature is higher during the dry periods than wet, and thus, I speculate that BVOC emissions are higher during the dry period. Quantitative information and reasons for the increase of HC emissions should be given.

Response: Thank the reviewer for this question. The sentence has been revised as follows.

"During the wet period, the concentration of anthropogenic HC increased".

As the reviewer mentioned, the variation of the flux of the biogenic hydrocarbons depends on the geological factors and the metrological conditions mainly influenced by temperature and sunlight intensity. The SOA model simulation result shows the gradual increase of biogenic SOA mass from early May to early June due to seasonal change under the metrological conditions. There are some variations in the biogenic SOA production due to the daily change in the biogenic hydrocarbon emission associated with differences in temperature and sunlight intensity. However, the variation of the biogenic SOA mass is relatively small compared to that of anthropogenic origin SOA during the simulated period as seen in Figure 7.

- L308: "isoprene SOA is negligible at all sites due to low isoprene emissions". Information of isoprene emissions (preferably with terpene and aromatics) is required.

Response: In South Korea, biogenic hydrocarbons mainly originate from coniferous tree, which is dominated by the pine trees. Therefore, the monoterpene flux in South Korea are relatively significant (Lee et al., 2017). An estimation of isoprene and monoterpenes emissions in the global scale based on 30-years Megan simulation showed that the relative significance of isoprene contribution to the biogenic hydrocarbons is little. In order to response to the reviewer, we added sentence to the revised manuscript (last paragraph in Section 3.3).

"An estimation of biogenic hydrocarbon emissions in the global scale, simulated by Sindelarova et al. (2014) by using Megan for 30 years, showed that the relative significance of isoprene emission is little in South Korea."

**2 Comments on precursors' contributions:**

You wrote in L308 as "Isoprene SOA is negligible at all sites", and concentrations of isoprene SOA was small over the domain as shown in Figure 8 (g) and (h). However, previous observational and simulation studies have indicated that isoprene SOA has important contributions in East Asia in May-June (e.g., Hu et al., Zhu et al., and Ding et al.). I recommend the authors to discuss the differences of your estimate with previous studies.

Hu et al. (2017) doi:10.5194/acp-17-77-2017

Zhu et al. (2018) doi: 10.1016/j.apr.2017.09.001

Ding et al. (2016) doi: 10.1038/srep20411

Response: The emission of isoprene is spatial sensitive. Due to the difference in tree species and climates (temperature, humidity, sunlight, and precipitation), the isoprene emission in Northeast Asia is much lower than that in Southeast Asia. The modeling results in the previous studies also reported (Ding et al., 2016; Hu et al., 2017; Zhu et al., 2018) that the isoprene emission was significantly lower in Northeast China than that in South China. South Korea has a clear four seasons with cold winter and hot summer. In addition, the emission of isoprene in May during the KORUS-AQ campaign is relatively lower than that in the summer seasons (July-September).

**3. Comments on OMH and OMP**

You wrote in L324-326 that "Under the dry period (Fig. 3), the predicted SOA mass by the UNIPAR model is dominated by gas-particle partitioning onto organic phase and oligomerization in organic aerosol. During the wet period, SOA production forms mainly through gas-aqueous partitioning and aqueous reactions."

I could not get how did you separate contributions of oligomer SOA and SOA from aqueous-phase reactions (I guess both are categorized OMH). Quantitative information of the contributions of the three pathways is helpful to readers.

Response: In the UNIPAR model, the SOA formation is processed by the two mechanisms: OMP from multiphase partitioning and OMH from oligomerization in both organic and salted aqueous solution. To clarify the SOA formation in the UNIPAR model, the following sentence is added to the revised manuscript and reads now (item 4 and 7 in section 2.2).

"The SOA mass formed from the partitioning process (OMP) is attributed to Cor and Cin."

"The SOA mass in the UNIPAR model is attributed to OMP and OMH."

**4 Comment on OM and OC**

It is not clear whether you showed organic mass (OM) or organic carbon (OC) in Figures 3-5. I guess OM concentration is calculated by your simulation model, whereas OC concentration is measured by carbon analyzers. Conversion factor from OC to OM (or vice versa) should be explicitly noted.

Response: We thank the reviewer for this comment. Both observation data and simulation results are organic matter (OM). The figure captions in Figures 3 and 4 are corrected in the revised manuscript and reads now.

"Figure 3: Time profiles of OM concentration  $(\mu g/m^3)$  averaged over eight hours for the observation data and the CAMx simulation results at the (a) Bulkwang, (b) Olympic Park, (c) Daejeon, and (d) Gwangju supersites."

"Figure 4: Time profiles of hourly averaged OM concentrations ( $\mu g/m^3$ ) for the observation data and the CAMx simulation results at the (a) Bulkwang, (b) Olympic Park, (c) Daejeon, and (d) Gwangju supersites."

**Specific comments:**

L51: References for the following sentence is necessary: "In particular, the current model applied to regional scales suffers from a substantial negative bias under high humidity conditions."

Response: Reference has been added to the second paragraph in introduction of the revised manuscript and reads now.

"In particular, the current model applied to regional scales suffers from a substantial negative bias under high humidity conditions (Heald et al., 2011; Pye et al., 2017; Li et al., 2020)."

-L104: eight aromatics?

Response: Total 10 aromatics were included in the UNIPAR simulation: benzene, toluene, ethylbenzene, propylbenzene, o-xylene, m-xylene, p-xylene, 1,2,3-trimethylbenzene, 1,2,4-trimethylbenzene, and 1,3,5-trimethylbenzene. The description in the item 2 of the Section 2.2 has been updated and reads now.

"The UNIPAR model of this study includes 151 lumping species, of which 50 originate from ten aromatics (benzene, toluene, ethylbenzene, propylbenzene, o-xylene, m-xylene, p-xylene, 1,2,3-trimethylbenzene, 1,2,4-trimethylbenzene, and 1,3,5-trimethylbenzene);"

L214: VCPs sourced from "residential, commercial, and industrial sectors"?

Response: The sentence has been revised in the second paragraph of Section 3.1 and reads now.

"The SOA simulation needs to be updated to include sesquiterpenes, intermediate VOCs, and volatile chemical species sourced from residential, commercial, and industrial sectors (Mcdonald et al., 2018)."

L300: "OMH attributes to 50% of aromatic SOA": it appears OMH contribution is smaller than 50% in Fig. 7 (during the wet period).

Response: The sentence has been updated in the third paragraph of the Section 3.3 and reads now.

"In Fig. 7a-7d (UNIPAR), OMH attributes to 22% to 48% of aromatic SOA, showing the importance of heterogeneous reactions of aromatic products to form SOA during the KORUS-AQ campaign."

L342: 53% of total anthropogenic VOC emissions in LA?

Response: The sentence has been updated in the third paragraph in Section 4 and reads now.

"In addition, the recent study by Mcdonald et al. (2018) showed that volatile chemical products (>53% of total anthropogenic VOC emissions in Los Angeles, USA) originating from consumer and industrial products, which are currently unaccounted for in models, can significantly contribute to SOA burden in the urban atmosphere."

**Reference**

- Ding, X., He, Q.-F., Shen, R.-Q., Yu, Q.-Q., Zhang, Y.-Q., Xin, J.-Y., Wen, T.-X., and Wang, X.-M.: Spatial and seasonal variations of isoprene secondary organic aerosol in China: Significant impact of biomass burning during winter, Sci Rep-Uk, 6, 20411, 10.1038/srep20411, 2016.
- Heald, C. L., Coe, H., Jimenez, J. L., Weber, R. J., Bahreini, R., Middlebrook, A. M., Russell, L. M., Jolleys, M., Fu, T. M., Allan, J. D., Bower, K. N., Capes, G., Crosier, J., Morgan, W. T., Robinson, N. H., Williams, P. I., Cubison, M. J., DeCarlo, P. F., and Dunlea, E. J.: Exploring the vertical profile of atmospheric organic aerosol: comparing 17 aircraft field campaigns with a global model, Atmos. Chem. Phys., 11, 12673-12696, 10.5194/acp-11-12673-2011, 2011.
- Hu, J. L., Wang, P., Ying, Q., Zhang, H. L., Chen, J. J., Ge, X. L., Li, X. H., Jiang, J. K., Wang, S. X., Zhang, J., Zhao, Y., and Zhang, Y. Y.: Modeling biogenic and anthropogenic secondary organic aerosol in China, Atmos Chem Phys, 17, 77-92, 10.5194/acp-17-77-2017, 2017.
- Lee, J., Cho, K. S., Jeon, Y., Kim, J. B., Lim, Y.-r., Lee, K., and Lee, I.-S.: Characteristics and distribution of terpenes in South Korean forests, Journal of Ecology and Environment, 41, 19, 10.1186/s41610-017-0038-z, 2017.
- Li, J., Zhang, H., Ying, Q., Wu, Z., Zhang, Y., Wang, X., Li, X., Sun, Y., Hu, M., Zhang, Y., and Hu, J.: Impacts of water partitioning and polarity of organic compounds on secondary organic aerosol over eastern China, Atmos. Chem. Phys., 20, 7291-7306, 10.5194/acp-20-7291-2020, 2020.
- McDonald, B. C., de Gouw, J. A., Gilman, J. B., Jathar, S. H., Akherati, A., Cappa, C. D., Jimenez, J. L., Lee-Taylor, J., Hayes, P. L., and McKeen, S. A.: Volatile chemical products emerging as largest petrochemical source of urban organic emissions, Science, 359, 760-764, 2018.
- Pye, H. O. T., Murphy, B. N., Xu, L., Ng, N. L., Carlton, A. G., Guo, H. Y., Weber, R., Vasilakos, P., Appel, K. W., Budisulistiorini, S. H., Surratt, J. D., Nenes, A., Hu, W. W., Jimenez, J. L., Isaacman-VanWertz, G., Misztal, P. K., and Goldstein, A. H.: On the implications of aerosol liquid water and phase separation for organic aerosol mass, Atmos Chem Phys, 17, 343-369, 10.5194/acp-17-343-2017, 2017.
- Sindelarova, K., Granier, C., Bouarar, I., Guenther, A., Tilmes, S., Stavrakou, T., Müller, J. F., Kuhn, U., Stefani, P., and Knorr, W.: Global data set of biogenic VOC emissions calculated by the MEGAN model over the last 30 years, Atmos. Chem. Phys., 14, 9317-9341, 10.5194/acp-14-9317-2014, 2014.
- Zhou, C., Jang, M., and Yu, Z.: Simulation of SOA Formation from the Photooxidation of Monoalkylbenzenes in the Presence of Aqueous Aerosols Containing Electrolytes under Various NOx Levels, Atmos. Chem. Phys., 19, 5719-5735, 2019.
- Zhu, W., Luo, L., Cheng, Z., Yan, N., Lou, S., and Ma, Y.: Characteristics and contributions of biogenic secondary organic aerosol tracers to PM2.5 in Shanghai, China, Atmos Pollut Res, 9, 179-188, https://doi.org/10.1016/j.apr.2017.09.001, 2018.

---

## Author Comment (AC2)

**Manuscript #: acp-2021-1002**

**Response to Anonymous Referee #RC2:**

"Yu et al., describe the impact of implementing a state-of-science module for the formation of secondary organic aerosol from traditional as well as "novel" pathways including multi-phase processes involving particles. They evaluate their model against ground observations taken during a recent field campaign over South Korea for the duration of 1 month.

The manuscript is well written and presents the main findings in a concise and understandable fashion. Conclusions are sound presented in a balanced manner, mostly considering the state of the science in the field at this time. My main points are (1) the need to also focus on the remainder of the lifecycle of organic matter in the atmosphere, (2) to make better use of the wealth of data generated during KORUS-AQ to evaluate the model, and (3) a broader evaluation of the model performance. I would recommend major revisions."

Response: We appreciate the reviewer for the time and effort on this study. Additional discussion about the aerosol lifecycle and the model evaluation using the field data are added in the revised manuscript. A line-by-line response to the reviewer's comment is listed as below.

**Main points:**

(1) Organic aerosol lifecycle

Concentrations of OA in the atmosphere are determined by its sources (emission, secondary production) as well as its sinks. The authors claim to do better firstly because their model represents more of the physics and chemistry that probably takes place in the atmosphere, and secondly because it evaluates better against observations. I concur with the former, but find the latter needs to be discussed (further) in the manuscript. A lot of work has shown that OA can photolysis, age, and deposit in ways most models do not consider, thereby changing its properties and lifetime. Why is being closer to observations now "better" with UNIPAR, maybe you are just compensating model deficiencies in other areas?

Response: We agree with the reviewer that there are uncertainties associated with aerosol aging in the model of this study. In the CAMx-UNIPAR simulation, the deposition of the OA originated SOA was treated based on the one size bin for the fine particulate matter. The UNIAPR model include the aging of gas products but needs an aerosol aging process due to OA aging in the future. The discussion about aerosol deposition can be found in the section 2.1. The discussion about OA aging and aerosol deposition has been added to the section 4 Atmospheric implications and uncertainties and reads now.

"In addition, the deposition of SOA was estimated with the one particle size bin. The different particle size can have different sink fluxes causing the uncertainty in the lifetime

of OM. The UNIPAR model is capable of predicting aging of gas products but currently has no feature for OM aging."

(2) KORUS-AQ campaign data

KORUS-AQ was also a large aircraft campaign, a treasure trove of observations is readily available (including OA data!) from several aircraft platforms. It would be almost negligent to not use this data to evaluate a 3D m model simulation and instead focus only on three ground stations. There is so much more to learn about OA model performance when looking "up in the sky"!

Response: Thank the reviewer for the important comments. In the future, we will utilize the aircraft data. Prior to the compare the simulation with aircraft data, the evaluation of the aircraft data based on emission sources need to be performed.

(3) Model performance evaluation

The authors have provided quite some data to look at overall model performance, but I suggest completing this in the following areas: how well is $NO_x$ represented, what is the performance for temperature and humidity, and how well does the model represent the main SOA precursor levels (aromatics, terpenes and isoprene)? Again, see point 2, there is a wealth of data available!

Response: The $NO_x$ observations during the KOROS-AQ were not available in many sites. We can find $NO_x$ observation data only at the Olympic Supersite of the selected four sites of this study (Figure S4). The sites chosen in this study are at best in the availability of various data and timeline.

The temperature and humidity were produced from the WRF model and they accorded well with the filed measurements ($R^2$ = 0.9999 for temperature and $R^2$=0.9688 for relative humidity). This information can be found in the figure caption in Figures S5 and S6 and reads now:

"The temperature inputs from the WRF simulation accords well with the filed measurements ($R^2$ = 0.9999)."

"The RH for the CAMx meteorological inputs from the WRF simulation accords well with the filed measurements ($R^2$ = 0.9688)."

Specific comments:

15 ff "explicit" gas-phase chemistry?

Response: Corrected and reads now.

"The UNIfied Partitioning-Aerosol phase Reaction (UNIPAR) model utilizes the explicit gas mechanism to better predict SOA formation from multiphase reactions of hydrocarbons."

37 why italic for "via"?

Response: Word "via" was not italicized now.

37 HC abbreviation, first mention, explain!

Response: This has been corrected in the 2nd paragraph in introduction.

48: The fact that SOA precursors can undergo multi-phase chemistry involving a liquid-phase implies they are hygroscopic, which leads to important questions regarding their fate in the atmosphere. E.g., is deposition accounted for correctly (see, e.g., Knote et al., 2015)? Also, given that at least during daytime, we are in a photochemically active environment, what about photolysis losses of oxidized volatile organic compounds (OVOCs) (e.g., Hodzic et al, 2015)?

Knote, C., Hodzic, A., and Jimenez, J. L.: The effect of dry and wet deposition of condensable vapors on secondary organic aerosols concentrations over the continental US, Atmos. Chem. Phys., 15, 1–18, https://doi.org/10.5194/acp-15-1-2015, 2015.

Hodzic, A., Kasibhatla, P. S., Jo, D. S., Cappa, C. D., Jimenez, J. L., Madronich, S., and Park, R. J.: Rethinking the global secondary organic aerosol (SOA) budget: stronger production, faster removal, shorter lifetime, Atmos. Chem. Phys., 16, 7917–7941, https://doi.org/10.5194/acp-16-7917-2016, 2016.

Response: In the regional scale model, the produced SOA sinks via the dry and the wet deposition. The UNIPAR model considers the dynamic oxidation of gas products under varying $NO_x$ levels. In addition, the lumping species also sink to aqueous droplets (i.e., cloud) in the model and lose via the dry deposition. We agree with the reviewer in that SOA can be decomposed due to photolysis under the sunlight. The discussion about OA aging and the aerosol deposition has been added to the section 4, Atmospheric implications and uncertainties, and reads now.

"In addition, the deposition of SOA was estimated with the one size bin for the fine particulate matter. The different particle size can have different sink flux causing the uncertainty in the aerosol lifecycle of OM. The UNIPAR model is capable of the prediction of gas products aging but currently has no feature for OM aging."

49: citations are for box models, better suited in relation to this study are examples for the regional and global scale, e.g. Budisulistiorini et al., 2017 (IEPOX), Knote et al., 2015 (GLYOXAL) and Stadler et al., 2018 (IEPOX), Myriokefalitakis et al., 2008 (GLYOXAL), respectively

Sri Hapsari Budisulistiorini, Athanasios Nenes, Annmarie G. Carlton, Jason D. Surratt, V. Faye McNeill, and Havala O. T. Pye Environmental Science & Technology 2017 51 (9), 5026-5034 DOI: 10.1021/acs.est.6b05750

Knote, C., Hodzic, A., Jimenez, J. L., Volkamer, R., Orlando, J. J., Baidar, S., Brioude, J., Fast, J., Gentner, D. R., Goldstein, A. H., Hayes, P. L., Knighton, W. B., Oetjen, H., Setyan, A., Stark, H., Thalman, R., Tyndall, G., Washenfelder, R., Waxman, E., and Zhang, Q.: Simulation of semi-explicit mechanisms of SOA formation from glyoxal in aerosol in a 3-D model, Atmos. Chem. Phys., 14, 6213–6239, https://doi.org/10.5194/acp-14-6213-2014, 2014.

Stadtler, S., Kühn, T., Schröder, S., Taraborrelli, D., Schultz, M. G., and Kokkola, H.: Isoprene-derived secondary organic aerosol in the global aerosol–chemistry–climate model ECHAM6.3.0–HAM2.3–MOZ1.0, Geosci. Model Dev., 11, 3235–3260, https://doi.org/10.5194/gmd-11-3235-2018, 2018.

Myriokefalitakis, S., Vrekoussis, M., Tsigaridis, K., Wittrock, F., Richter, A., Brühl, C., Volkamer, R., Burrows, J. P., and Kanakidou, M.: The influence of natural and anthropogenic secondary sources on the glyoxal global distribution, Atmos. Chem. Phys., 8, 4965–4981, https://doi.org/10.5194/acp-8-4965-2008, 2008.

Response: In order to respond to the reviewer, several references have been added to the revised manuscript and reads now (the 2nd paragraph in introduction).

"Several chemical transport models account for the aqueous reactions of few explicit products (i.e., glyoxal and IEPOX (epoxy diols form isoprene products)) that potentially may significantly impact the SOA formation (Ervens et al., 2011; Sumner et al., 2014; Budisulistiorini et al., 2017; Knote et al., 2014)."

51: citation to prove this claim?

Response: The references have been added to the revised manuscript and reads now (the 2nd paragraph in introduction).

"In particular, the current model applied to regional scales suffers from a substantial negative bias under high humidity conditions (Heald et al., 2011; Pye et al., 2017; Li et al., 2020)."

52: which "conventional model", not true in this broad claim form!

Response: The sentence has been modified in the revised manuscript and reads now (the 2nd paragraph in the introduction).

"The SOA model, such as the partitioning-base two product model, has no feature for SOA formation via aqueous phase reactions of different oxygenated products formed from various HCs."

56: all these citations are the reference for UNIPAR, or is there a single one that serves as reference? It needs to be made clear where UNIPAR is scientifically published.

Response: The sentence has been modified in the revised manuscript and reads now.

"The UNIfied Partitioning-Aerosol phase Reaction (UNIPAR) model was developed by Im et al. (2014) to predict SOA mass based on multiphase reactions of toluene and 1,3,5-trimethylbenzene, was developed. In the UNIPAR model, the products predicted using explicit gas mechanisms are lumped based on volatility and emerging chemistry in the aerosol phase. This UNIPAR model has been extended to various SOA originating from isoprene, terpenes, aromatics, and gasoline and demonstrated through the extensive photochemical outdoor smog chamber data (Beardsley and Jang, 2016; Cao and Jang, 2010; Zhou et al., 2019; Yu et al., 2021; Han and Jang, 2022)."

59: what is "arrayed" supposed to mean?

Response: Word "arrayed" has been changed to "estimated" (the 3$^{rd}$ paragraph in introduction).

"The model parameters linked to the thermodynamic properties and aerosol chemistry are also estimated according to the lumped species characteristics."

62: CAMx needs to be introduced (regional scale model...) and cited!

Response: This sentence has been modified in the revised manuscript and reads now (the 4$^{th}$ paragraph in introduction).

"In this study, the UNIPAR model was incorporated with the CAMx model (comprehensive air quality model with extensions, v7.10) (Environ, 2020) to predict the SOA formation in the regional scale during the Korean-United States Air Quality (KORUS-AQ) campaign that took place between 10 May, 2016 and 10 June, 2016."

75: SOAP is quite outdated - there should be more recent developments for CAMx that would better show the effect of UNIPAR over the _current_ state of science. See e.g. Jiang et al., 2021, for references.

Jiang, J., El Haddad, I., Aksoyoglu, S., Stefenelli, G., Bertrand, A., Marchand, N., Canonaco, F., Petit, J.-E., Favez, O., Gilardoni, S., Baltensperger, U., and Prévôt, A. S. H.: Influence of biomass burning vapor wall loss correction on modeling organic aerosols in Europe by CAMx v6.50, Geosci. Model Dev., 14, 1681–1697, https://doi.org/10.5194/gmd-14-1681-2021, 2021.

Response: Thank you for the suggestion. In the future, we will implement the updated modules and parameters. In this study, our focus is the demonstration of the importance of aqueous phase reactions of organic species to form SOA by suing the UNIPAR model.

75: Also, how do comparable model systems fare during KORUS-AQ? There is a good overview by Park et a., 2021, on multi-model results that should provide insights into how the model used here fares compared to others.

Rokjin J. Park, Yujin J. Oak, Louisa K. Emmons, Cheol-Hee Kim, Gabriele G. Pfister, Gregory R. Carmichael, Pablo E. Saide, Seog-Yeon Cho, Soontae Kim, Jung-Hun Woo, James H. Crawford, Benjamin Gaubert, Hyo-Jung Lee, Shin-Young Park, Yu-Jin Jo, Meng Gao, Beiming Tang, Charles O. Stanier, Sung Soo Shin, Hyeon Yeong Park, Changhan Bae, Eunhye Kim; Multi-model intercomparisons of air quality simulations for the KORUS-AQ campaign. Elementa: Science of the Anthropocene 21 January 2021; 9 (1): 00139. doi: https://doi.org/10.1525/elementa.2021.00139

Response: The focus of our study is to demonstrate the importance of multiphase partitioning of organic species and their aqueous reactions (151 lumping species) by using the UNIPAR model. In particular, the UNIPAR can estimate the activity coefficient of lumping species on aqueous phase allowing the impact of aerosol water mass on SOA formation. In addition, the simulation in this study was compared to the ground-based observations during KORUS-AQ campagin.

In order to respond to the reviewer, we discussed the recent model simulation of the organic aerosol in the regional scale in the second paragraph of introduction and in the last paragraph of Section 3.1

"Park et al. (2021) extensively evaluated the prediction of the organic aerosol produced during the KORUS-AQ campaign by using different air quality models, which were varying in chemistry mechanisms, aerosol thermodynamics, the types of SOA precursors, and the SOA schemes. In their study, the SOA formation was simulated with the SOAP, the 4 bin-base VBS or the 5-bin-base VBS modules. The predicted organic aerosol masses were, however, underestimated compared to observation data (HR-ToF-AMS) suggesting the limitation of the current SOA modules."

"For organic matter, the average Normalized Mean Bias (NMB, %) between model predictions and observations at the four monitoring sites are -50% for CAMx-SOAP and -39% for CMAx-UNIPAR. A similar level of the NMB ($\approx$ 46%) was reported in the previous simulation for the same campaign (Park et al., 2021)"

141 ff: are organic acids considered when calculating aerosol acidity? How good is your aerosol water content, as it is crucial for acidity calculations?

Response: The aerosol acidity and the aerosol water content both were estimated by using the ISORROPIA inorganic thermodynamic model. In general, many inorganic thermodynamic models use the ZSR relation to estimate water activity of the system that is directly related to predict aerosol water content (Stokes and Robinson, 1966; Zdanovskii, 1948). It is known that the estimation of water prediction is relatively accurate and similar between models. However, the calculation of the activity coefficient of the proton in the highly concentrated salted system are uncertain due to the lack of database and it is various

between models as discussed in the previous studies (Jang et al., 2020; Pye et al., 2020). During the KORUS-AQ campaign, inorganic acids were significantly titrated, and aerosol acidity was near neutral. Thus, the aerosol water mass mainly influenced aqueous phase reactions of organics and their partitioning to aqueous phase.

142: typo "ISORRIPIA"

Response: This has been corrected.

"In order to process SOA formation in the inorganic aqueous phase, the inorganic composition and aerosol acidity are predicted by using the inorganic thermodynamic model, ISORROPIA (Fountoukis and Nenes, 2007), and then incorporated into the UNIPAR model. For the ISORROPIA model, mutual deliquescence relative humidity (MDRH) is predicted."

155: "MOZART", all caps

Response: This has been corrected.

"The boundary conditions were converted from the MOZART-4 global simulation results (https://www.acom.ucar.edu/wrf-chem/mozart.shtml) (Emmons et al., 2010)."

194: I would expect at least a short model evaluation for the main drivers of OA formation: meteorology (temperature, humidity, radiation), oxidants ($O_3$, $NO_x$) and precursors (aromatics, terpenes, isoprene). See also main concerns.

Response: Please find the response to the 3$^{rd}$ main comment above.

210ff: how well does your model capture the precursors you actually included? Measurements of aromatics, terpenes and isoprene should be available!

Response: As seen in Section 2.3.2, the emission of air pollutants was determined by using SMOKE from emission inventories originating from various sources (i.e., point sources, area sources, biogenic sources (MEGAN), automobiles non-mobile sources, etc). During the KORUS-AQ campaign, only few precursors were monitored (i.e., toluene). The predicted toluene was on average 94% of observed toluene.

356ff: "Furthermore, the UNIPAR model integrated with regional models enables better prediction of future SOA burdens under different scenarios of air pollutant emissions." This statement is too broad to be supported by the analysis shown here - why are you better equipped represent future scenarios better? Because you seem to compare better to 3 ground stations in one geographical corner of the world for 1 month in one year? Because you represent processes better? Address!

Response: The sentence has been removed.

Figure S5: do model and measurements coincide (i.e., the model is perfect), or might there be a difference in modelled vs. measured temperature, leading to differences in the thermodynamic environment that should be discussed? Figure S6: same question as for S6!

Response: Temperature and humidity in the model were obtained from the WRF simulation results and they accorded well with the measurements ($R^2$=0.999 for temperature and $R^2$=0.969 for relative humidity). This information can be found in the figure captions in Figures S5 and S6 in the revised manuscript.

---

## Author Response (AR2)

Manuscript #: acp-2021-1002

**Response to Anonymous Referee #2:**

We would like to thank reviewer for the time and the constructive comments on our work. The comments are reproduced below along with the author response in red color and any significant changes made to the manuscript or supporting information in blue color.

**Comment #1:** Regarding the OA lifecycle:

(1) Deposition of particulate OA: using one size bin to estimate deposition fluxes for all sizes is overly simplified and will introduce an error, as the authors correctly state now. Why authors choose to implement it in such a way is unclear to me, as it would not introduce considerable computational overhead to do this for all sizes.

Response: The CAMx regional model employs a static two-mode coarse/fine (CF) scheme for the particle mass distribution. As an option, the evolving multi-section size scheme (Carnegie Mellon University, CMU) can be used but its compatibility with other model components is limited. For example, the current CAMx allows to operate the CMU scheme with ISORROPIA and SOAP chemistry integrated with the CB05 gas mechanism, but its development is not available for other gas mechanisms (i.e., SAPRC) or SOA modules (VBS). To clarify the setup of the aerosol size bin, the sentence has been added to the section "2.3.1 Simulation domain and model configurations" in the revised manuscript and reads now.

"The two-mode coarse/fine (CF) scheme for the particle mass distribution was employed. In the CAMx, the multi-section size scheme can be operated with ISORROPIA and SOAP chemistry integrated with CB05 but it is currently not comparable with other gas mechanisms such as SAPRC or other SOA modules (i.e., VBS modules)."

(2) Deposition of gaseous OA (also known as condensable vapors): there has been quite some work on the influence of the wet and dry deposition of gaseous-phase OA components (condensable vapors) have also on particulate OA due to the fact that they are in thermodynamic equilibrium. How are condensable vapors deposited, what assumptions about Henrys law constants are made, and are they similar for SOAP and UNIPAR?

While you have somewhat answered my question in the authors response, none of the gas-phase deposition or photolysis language actually made it into the manuscript. You should revise your changes. I think these merits mentioning it also to the reader of manuscript if it is accepted.

Response: The deposition of aerosols and gas species in this study was estimated with the default module existing in the CAMx model and has no difference between SOAP and UNIPAR simulations.

As described in the CAMx User Guide v7.10 (Environ, 2020), the wet deposition model in the CAMx regional model employs a scavenging coefficient, which are determined differently for gases and particles. Briefly, the wet scavenging is calculated for each layer within the precipitating grid column from the top of the precipitation profile to the surface. The gases species dissolved in the precipitation water are in equilibrium with ambient air concentrations according to Henry's law constant, aqueous dissociation, cloud water temperature and acidity. In this study, the Henry's law constant of gas species used in the gas oxidation mechanism, SAPRC07TC, was preset in the parameter files of the CAMx regional model and used for both SOAP and UNIPAR simulations. For particles, the wet scavenging rate is dependent on the rainfall rate, the drop diameter, and the collection efficiency, which is a complex function of particle size, density, hydrometeor size, fall speed, kinematic viscosity of air and water. The dry deposition of gases compound and particles are estimated depending on a deposition velocity, which is function of solubility, diffusivity, density, particle size, the meteorology, and the surface characteristics.

To explain the aerosol and gas deposition process in CAMx, the following sentence has been added to the section "2.3.1 Simulation domain and model configurations" and reads now.

"The dry and wet depositions of aerosols and gas species were estimated with the module existing in the CAMx model for both SOAP and UNIPAR simulations. The detailed explanation for the deposition model can be found in the CAMx User Guide v7.10 (Environ, 2020)."

**Comment #2:** Regarding the aircraft data from KORUS-AQ:

I would strongly suggest to the authors to reconsider their stance on (not doing) a model evaluation against KORUS-AQ aircraft data. Organic aerosol, created from a multitude of sources and processes, constantly changing in the atmosphere, and existing in thermodynamic equilibrium between particle and gas-phase is such a complex system that special care needs to be taken to evaluate an equally complex model parameterization. Authors would be well advised to evaluate their model under as many diverse atmospheric conditions (concentration levels, distance from emission source, mixture types, temperatures, humidities) as they can get their hands on to get a robust understanding of the actual performance of their parameterization. Using only a few ground stations which are in principle prone to local processes not resolved by the model grid (strong emitters close by, special topography and wind systems, nocturnal boundary layer height ...) is not

enough. Authors could also not dispel my concerns regarding compensating errors from looking only towards OA formation instead of all processes in the OA lifecycle, which casts further doubt on evaluating only against a few ground stations.

And finally, I do not understand what authors meant to say by "Prior to the compare the simulation with aircraft data, the evaluation of the aircraft data based on emission sources need to be performed.". The aircraft data collected during KORUS-AQ has been quality controlled and evaluated multiple times in diverse contexts, see the KORUS-AQ overview paper (Crawford et al., 2021, https://doi.org/10.1525/elementa.2020.00163) and references therein.

Response: The aircraft data that are collected within several hours is different from ground-based data collected in a continuous mode. Aircraft data collection is also performed under the fast speed (several kilometers per minute) and its altitude ranges from several hundred meter to 7km (sometimes, higher than Planetary Boundary Layer (PBL)). Aircraft data is influenced by vertical convection and mixing of an air parcel. For understanding of aerosol compositions and chemical transformation, both aircraft data and ground-based observations are valuable. The flight tracks of NASA DC-8 aircraft missions during the simulated period of this study are shown in Figure S11 in the revised supporting information.

[Figure]

**Figure S11**. Flight tracks of NASA DC-8 aircraft missions during the simulated period and regions of this study.

We have reviewed the available data of NASA DC-8 aircraft mission during the KORUS-AQ campaign. The observed data for ozone, NO, $NO_x$, and toluene were available in 12 different flight missions (4-6 hours each) between May 10 and June 10 in 2016. The comparison of the observations and the model predictions for ozone, NO, $NO_x$, and toluene gas is plotted in Figure S11 in the revised supporting information. The AMS data collected from the aircraft during the DC-8 flight missions are Organic Carbon (OC) concentrations specifically in $PM_1$ and thus, they are not directly comparable to the simulated total organic matter. Figure S12 shows the correlation between AMS data and the simulated primary organic aerosol (POA) or the simulated secondary organic aerosol (SOA). A higher

correlation coefficient appears between AMS data and the simulated SOA (PCC = 0.57) than that between AMS data and the simulated POA (PCC = 0.38), indicating that the observed OC in the high altitude is much more influenced by secondary pollutants.

Following paragraph is added to the end of Section 2.4 "Observations during the KORUS-AQ campaign" in the revised manuscript and reads now.

"The KORUS-AQ campaign performed several flight measurements by using the NASA DC-8 research aircraft with a comprehensive payload for in situ sampling of trace gas and aerosol compositions. Fig. S11 shows the flight tracks of the NASA DC-8 aircraft missions during the KORUS-AQ campaign between May 10 and June 10 in 2016. The observed airborne concentrations of ozone, NO, $NO_2$, and toluene are plotted against the simulation from the CAMx-UNIPAR model (Fig. S12)."

[Figure]

Figure S12. The observations vs. the simulated concentration (ppb) of (a) ozone, (b) NO, (c) NO$_2$, and (d) toluene during the NASA DC-8 aircraft missions of the KORUS-AQ campaign. The CAMx-UNIPAR was used for the simulation output. Terms "MBE", "PCC", and "NMB" represent mean bias error, Pearson correlation coefficient, and normalized mean bias, respectively. The detailed equations for the statistic calculation are listed in Table S2. The grid size for CAMx-UNIPAR simulation was 9 km × 9 km. The data collection in on-board observation was performed every second. The on-board data was averaged for 30 seconds, which is equivalent to approximate 6 km distance (less than a grid width). The maximum aircraft ground speed was about 200 m/s.

[Figure]

Figure S13. The correlation between the observed organic aerosol (OA) concentration (ug/m$^3$) and the predicted primary organic aerosol (POA) (a) or the predicted of SOA concentration (µg/m$^3$) (b). The observed OA data in PM$_1$ were measured by using Aerosol Mass Spectrometer (AMS). Term "PCC" is the Pearson correlation coefficient. The simulated SOA mass is the sum of the OM produced via gas-particle partitioning and heterogeneous reactions of organics by using the UNIPAR module.

The section for gas simulation was newly added to "3.4 Simulated concentrations of gaseous species" in the revised manuscript and reads now.

"**3.4 Simulated concentrations of gaseous species**

Fig. S10 illustrates the correlation between the 8-hour averaged observations and the 8-hour averaged predictions of ozone, NO$_x$, SO$_2$ and toluene at the Olympic Park supersite. In general, the model prediction slightly underestimates ozone (Fig. S10a), SO$_2$ (Fig. S10c), and toluene (Fig. S10d), but overestimates NO$_x$ (Fig. S10b). Similarly, underestimation of ozone appeared in the on-board data (Fig. S12a). This underestimation

could be explained by the missing or the underestimation of ozone precursors (i.e., toluene as shown in Fig. S12d) in the current emission inventories.

Fig. S13 shows the correlation between Aerosol Mass Spectrometer (AMS) data and the simulated primary organic aerosol (POA) or the simulated secondary organic aerosol (SOA). A higher correlation coefficient appears between the AMS data collected during the DC-8 flight missions and the simulated SOA (PCC = 0.57) than that between AMS data and the simulated POA (PCC = 0.38), indicating that observed OC in the high altitude is more influenced by secondary pollutants."

**Comment #3:** Regarding model performance evaluation:

Authors did not reply to my comment on evaluating against main SOA precursors - this is crucial to understand whether their OA formation processes are accurate. Again - there is data readily available.

Authors state regarding temperature and RH evaluation: "The temperature inputs from the WRF simulation accords well with the filed

88 measurements (R2 = 0.9999)." and "The RH for the CAMx meteorological inputs from the WRF simulation accords well with the

94 filed measurements (R2 = 0.9688)." None of the plots in S5 or S6 show a line for observed values, and I find it difficult to believe they would match perfectly all the time, and a R2 (squared!) of 0.9999 is not believable. Also, it is unclear to me what "RH for the CAMx meteorological inputs from the WRF simulation" is supposed to mean. Finally, there is a typo in "filed" measurements.

Response: In the previous revision, the intercept in the statistic regression between the observations and the simulation for temperature and RH were set to zero, which resulted $R^2$ value as high as 0.99. When the intercept is included, the $R^2$ is 0.885 for temperature and 0.738 for relative humidity. The plots for the observations versus the model prediction from WRF are newly added to the Supporting information (Figure S8). The predicted temperature at the Olympic Park supersite has less bias (NMB=-0.002) from observations compare to RH (NMB=-0.03).

[Figure]

Figure S8. Observations versus WRF simulations for (a) temperature (K) and (b) relative humidity (RH) at the Olympic Park supersite. "MBE", "PCC", and "NMB" represent mean bias error, Pearson correlation coefficient, and normalized mean bias, respectively. The definitions for the statistic calculation are listed in Table S2.

To evaluate the model performance, the time profiles of the ground observations and simulation from CAMx-UNIPAR for toluene and benzene at the Olympic Park supersite are newly added to the supporting information (Figure S5). For isoprene, the observation was not available. The correlation between observations and simulations for ozone, $NO_x$ $SO_2$, and toluene at the Olympic Park supersite are shown in Figure S10 in the supporting information.

[Figure]

**Figure S5.** Time profiles of the observed and predicted concentrations of (a) 8-hour averaged toluene, (b) 8-hour averaged benzene, and (c) 8-hour averaged isoprene at the Olympic Park supersite. For isoprene, the observation was not available.

[Figure]

**Figure S10.** The 8-hour averaged observations vs. simulated concentration (ppb) of (a) ozone, (b) NOₓ, (c) SO₂, and (d) toluene at the Olympic Park supersite. The CAMx-UNIPAR was used for the simulation output. Terms "MBE", "PCC", and "NMB" represent mean bias error, Pearson correlation coefficient, and normalized mean bias, respectively.

The following sentences are added to the end of section 2.4, "Observations during the KORUS-AQ campaign" of the revised manuscript and read now.

"In Fig. S5, the simulated concentration of SOA precursors, including toluene, benzene and isoprene, are plotted against the observations at the Olympic Park supersite. For isoprene, the observation was not available. For meteorological inputs, the observed temperature and RH at the Olympic Super site are plotted versus the simulations in Fig. S7.

Overall, the smaller bias between observations and predictions appeared in temperature compared to RH."

**Comment #4:** Regarding organic acids and acidity calculations:

Thank you for this concise elaboration - I would strongly suggest to actually put it into the manuscript, so other readers will also benefit!

Response: We thank the reviewer for the suggestion. The descriptions of the aerosol acidity and the aerosol water content have been added to Section S2 of the revised supporting information and read now.

**Section S2: Prediction of aerosol inorganic composition and aerosol acidity**

Both the aerosol inorganic composition and aerosol acidity are estimated by using the ISORROPIA inorganic thermodynamic model (Nenes et al., 1998; Fountoukis and Nenes, 2007). In general, many thermodynamic models, such as E-AIM (Clegg et al., 1998) and ISORROPIA (Nenes et al., 1998; Fountoukis and Nenes, 2007), employ the ZSR relation to estimate water activity of the system that is directly related to the prediction of aerosol water content (Stokes and Robinson, 1966; Zdanovskii, 1948). It is known that the estimation of water prediction is relatively accurate and similar between models. However, the calculation of activity coefficient of the proton in the highly concentrated salted system are uncertain due to the lack of database and it is various between models as discussed in the previous studies (Jang et al., 2020; Pye et al., 2020). During the KORUS-AQ campaign, the inorganic acid was mostly titrated with ammonia gas as shown in Figure S1-S3, and aerosol was near neutral. Thus, the aerosol water mass mainly influenced aqueous phase reactions of organics and their partitioning to aqueous phase.

In order to accurately predict the aerosol phase status (liquid or solid phase), the prediction of deliquescence relative humidity (DRH, 0-1) and the efflorescence relative humidity (ERH, 0-1) are essential in this study. The mutual deliquescence relative humidity (MDRH) is predicted by using ISORROPIA model. In a multicomponent inorganic mixture, the MDRH is the RH that all salts are simultaneously saturated with respect to all components. ERH is predicted by using the neural network model based on inorganic composition (Yu et al., 2021) as following equations.

$$N_1 = 1.54463 \times f_{anion} - 0.9243 \times f_{nitrate} - 0.073745 \tag{S9}$$

$$N_2 = -0.63382 \times f_{anion} + 0.82856 \times f_{nitrate} + 0.288342 \tag{S10}$$

$$N_3 = -0.18594 \times f_{anion} + 0.63382 \times f_{nitrate} + 0.366726 \tag{S11}$$

$$N_i' = \begin{cases} N_i \geq 0, N_i \\ N_i < 0, 0 \end{cases} \quad i = 1,2,3 \tag{S12}$$

$$N_4 = -0.50581 \times N_1' - 1.15781 \times N_2' + 0.68805 \times N_3' + 0.33499 \qquad \text{(S13)}$$

$$N_4' = \begin{cases} N_4 \geq 0, N_4 \\ N_4 < 0, 0 \end{cases} \qquad \text{(S14)}$$

$$ERH = 2.21228 \times N_4' + 0.00018 \qquad \text{(S15)}$$

$f_{anion}$ is the fraction of anion charges to total ion charges excluding proton and $f_{nitrate}$ is the mole fraction of nitrate to total anion. Series of N in equations denotes nodes in the neural network model.

**Reference**

Clegg, S. L., Brimblecombe, P., and Wexler, A. S.: Thermodynamic Model of the System $H^+$-$NH_4^+$-$SO_4^{2-}$-$NO_3^-$-$H_2O$ at Tropospheric Temperatures, J Phys Chem A, 102, 2137-2154, DOI 10.1021/jp973042r, 1998.

ENVIRON, R.: User's Guide Comprehensive Air Quality Model with Extensions version 7.10, 2020.

Fountoukis, C. and Nenes, A.: ISORROPIA II: a Computationally Efficient Thermodynamic Equilibrium Model for $K^+$-$Ca^{2+}$-$Mg^{2+}$-$NH_4^+$-$Na^+$-$SO_4^{2-}$-$NO_3^-$-$Cl^-$-$H_2O$ Aerosols, Atmos Chem Phys, 7, 4639-4659, DOI 10.5194/acp-7-4639-2007, 2007.

Jang, M., Sun, S., Winslow, R., Han, S., and Yu, Z.: In situ aerosol acidity measurements using a UV–Visible micro-spectrometer and its application to the ambient air, Aerosol Sci Tech, 54, 446-461, 10.1080/02786826.2020.1711510, 2020.

Nenes, A., Pandis, S. N., and Pilinis, C.: ISORROPIA: A New Thermodynamic Equilibrium Model for Multiphase Multicomponent Inorganic Aerosols, Aquat Geochem, 4, 123-152, Doi 10.1023/A:1009604003981, 1998.

Pye, H. O. T., Nenes, A., Alexander, B., Ault, A. P., Barth, M. C., Clegg, S. L., Collett Jr, J. L., Fahey, K. M., Hennigan, C. J., Herrmann, H., Kanakidou, M., Kelly, J. T., Ku, I. T., McNeill, V. F., Riemer, N., Schaefer, T., Shi, G., Tilgner, A., Walker, J. T., Wang, T., Weber, R., Xing, J., Zaveri, R. A., and Zuend, A.: The acidity of atmospheric particles and clouds, Atmos. Chem. Phys., 20, 4809-4888, 10.5194/acp-20-4809-2020, 2020.

Stokes, R. and Robinson, R.: Interactions in aqueous nonelectrolyte solutions. I. Solute-solvent equilibria, The Journal of Physical Chemistry, 70, 2126-2131, 1966.

Yu, Z., Jang, M., and Madhu, A.: Prediction of Phase State of Secondary Organic Aerosol Internally Mixed with Aqueous Inorganic Salts, The Journal of Physical Chemistry A, 10.1021/acs.jpca.1c06773, 2021.

Zdanovskii, A.: New methods for calculating solubilities of electrolytes in multicomponent systems, Zh. Fiz. Khim, 22, 1478-1485, 1948.